# Riemannian stochastic optimization methods avoid strict saddle points

**Ya-Ping Hsieh,   Mohammad Reza Karimi,   Andreas Krause**
ETH Zürich
{yaping.hsieh, mkarimi}@inf.ethz.ch, krausea@ethz.ch

**Panayotis Mertikopoulos**
Univ. Grenoble Alpes, CNRS, Inria, Grenoble INP, LIG 38000 France
panayotis.mertikopoulos@imag.fr

## Abstract

Many modern machine learning applications – from online principal component analysis to covariance matrix identification and dictionary learning – can be formulated as minimization problems on Riemannian manifolds, typically solved with a Riemannian stochastic gradient method (or some variant thereof). However, in many cases of interest, the resulting minimization problem is *not* geodesically convex, so the convergence of the chosen solver to a desirable solution - i.e., a local minimizer - is by no means guaranteed. In this paper, we study precisely this question, that is, whether stochastic Riemannian optimization algorithms are guaranteed to avoid saddle points with probability 1. For generality, we study a family of retraction-based methods which, in addition to having a potentially much lower per-iteration cost relative to Riemannian gradient descent, include other widely used algorithms, such as natural policy gradient methods and mirror descent in ordinary convex spaces. In this general setting, we show that, under mild assumptions for the ambient manifold and the oracle providing gradient information, the policies under study avoid strict saddle points / submanifolds with probability 1, from any initial condition. This result provides an important sanity check for the use of gradient methods on manifolds as it shows that, almost always, the limit state of a stochastic Riemannian algorithm can only be a local minimizer.

## 1 Introduction

Modern machine learning systems have achieved remarkable success in the efficient optimization of highly non-convex functions using straightforward *Euclidean* techniques like stochastic gradient descent. A widely accepted hypothesis to explain this phenomenon is that, when the learning system under study – e.g., a neural network – is sufficiently expressive, local minimizers are essentially as good as global ones [21, 35]; by this token, a training algorithm can attain satisfactory performance by simply evading *saddle points* of the model's loss surface.

This observation has sparked a far-reaching research thread examining the behavior of various algorithms around saddle points in non-convex functions. Informally, these studies aim to tackle two fundamental questions:

$Q_1$: When does a given scheme, like stochastic gradient descent, avoid saddle points?

$Q_2$: Can we *augment* a given scheme so that it efficiently escapes saddle points?

Of the above questions, $Q_1$ focuses on *explaining* the empirical success of commonly used schemes, while the resolution of $Q_2$ usually revolves around proposing new schemes with desirable escape guarantees. These complementary perspectives have been extensively studied over the past decade,

37th Conference on Neural Information Processing Systems (NeurIPS 2023).

leading to a fairly complete understanding of how and when a Euclidean (stochastic) algorithm escapes saddle points, cf. [6, 28, 29, 41, 42, 46, 47, 53, 54] and references therein.

In parallel to the above, the recent surge of interest in Riemannian optimization has prompted a closer examination of *Riemannian* methods, thereby motivating an extension of $Q_1$ and $Q_2$ to a manifold setting – itself due to a wide range of breakthrough applications to machine learning and data science, from natural language processing, and signal processing to dictionary learning and robotics [48, 52, 64, 65]. As a result, there is an increasing demand for a comprehensive exploration of various spaces, such as the $d$-dimensional torus, Grassmannian or Stiefel manifolds, hyperbolic spaces, and many others.

Unfortunately, in a proper Riemannian setting, only $Q_2$ has received sufficient scrutiny thus far. Recent works by Criscitiello & Boumal [22] and Sun et al. [66] have shown that standard Riemannian deterministic algorithms can be augmented via the injection of an infinitesimal amount of noise (proportional to the method's desired accuracy), to achieve comparable escape guarantees in terms of oracle complexity as the corresponding Euclidean methods [30]. To the best of our knowledge, all existing results for $Q_1$ concern *deterministic* methods [28, 41, 42, 53] which are significantly limited in scope in large-scale machine learning applications because of their prohibitively high per-iteration cost.

**Our results and techniques.**   In view of the above, our paper aims to provide a general answer to $Q_1$ for a broad class of Riemannian stochastic optimization methods – including Riemannian stochastic gradient descent, its retraction-based and/or optimistic variants, etc. Concretely, we focus throughout on a flexible template of *Riemannian Robbins–Monro* (RRM) schemes [34, 59] which, in addition to the specific algorithms of interest mentioned above, also includes a range of Euclidean methods that can be analyzed efficiently from a Riemannian viewpoint.

Informally, our main result may be stated as follows:

> *Under any stochastic Riemannian Robbins–Monro method, the probability of*
> *converging to a strict saddle point* (*or a submanifold thereof*) *is zero.*

This statement provides firm grounds for accepting the output of a stochastic Riemannian optimization method as valid, as it shows that saddle points are avoided with probability 1.[1]  In the context of stochastic methods, our result builds on a series of foundational results by Pemantle [54] and Brandière & Duflo [20] who focused on *hyperbolic traps* (isolated saddle points with invertible Hessian). These results were subsequently extended by Benaïm & Hirsch [12] to a more general class of unstable *sets*, but this analysis remained grounded in a flat, Euclidean setting. The connecting tissue of our analysis with these works is the notion of an asymptotic pseudotrajectory (APT), which allows us to couple the long-run behavior of discrete-time RRM methods to that of an associated Riemannian gradient flow. [This discrete-to-continuous comparison is crucial for our analysis in order to apply center stable manifold techniques [63] to the RRM framework.] However, this comes at a significant cost, as establishing the APT property in a Riemannian setting is quite challenging. To achieve this, we employ a set of techniques recently developed by [33] which allow us to make this comparison precise and establish the desired avoidance result.

## 2   Background on Riemannian Manifolds

We begin with a brief overview of some basic definitions from Riemmanian geometry and optimization, solely intended to set notation and terminology; our presentation roughly follows the masterful account of Lee [43, 44], to which we refer the reader for a comprehensive introduction to the topic.

Le $\mathcal{M}$ be a $d$-dimensional, geodesically complete Riemannian manifold. Throughout the sequel, the tangent space to $\mathcal{M}$ at a point $x \in \mathcal{M}$ will be denoted by $\mathcal{T}_x\mathcal{M}$, and we will write $\dot{\gamma}(t) \in \mathcal{T}_{\gamma(t)}\mathcal{M}$ for the velocity vector to a smooth curve $\gamma \colon \mathbb{R} \to \mathcal{M}$ at time $t \in \mathbb{R}$. We will also write $\langle \cdot, \cdot \rangle_x$ for the metric at $x \in \mathcal{M}$, $\|\cdot\|_x$ for the associated norm, and $\mathrm{dist}(\cdot, \cdot)$ for the induced distance function on $\mathcal{M}$, the latter being defined via the minimization of the length functional $\mathcal{L}[\gamma] = \int \|\dot{\gamma}(t)\|_{\gamma(t)} \, dt$.

---

[1]We recall here that a strict saddle manifold is a set of critical points each of which has at least one negative Hessian eigenvalue. Such manifolds include ridge hypersurfaces and other connected sets of non-isolated saddle points that are common in the loss landscapes of high-dimensional machine learning models, so this result has significant cutting power in this regard.

Given a point $x \in \mathcal{M}$ and a tangent vector $z \in \mathcal{T}_x\mathcal{M}$, the (necessarily unique) geodesic emanating from $x$ along $z$ will be denoted by $\gamma_z$, and we define the exponential map at $x$ as $\exp_x(z) = \gamma_z(1)$ for all $z \in \mathcal{T}_x\mathcal{M}$ (recall here that $\mathcal{M}$ is assumed complete, so this map is well-defined for all $x \in \mathcal{M}$ and all $z \in \mathcal{T}_x\mathcal{M}$). Whenever well-defined, the inverse of $\exp_x$ will be written as $\log_x : \mathcal{M} \to \mathcal{T}_x\mathcal{M}$, with the understanding that the domain of $\log_x$ is actually the largest neighborhood of $x \in \mathcal{M}$ on which the restriction of $\exp_x$ is a (global) diffeomorphism; by definition, we have $\log_x(\exp_x(z)) = z$ for all $z$ for which the relevant quantities are well-defined. Finally, given a pair of points $x, x' \in \mathcal{M}$ and a tangent vector $z \in \mathcal{T}_x\mathcal{M}$, we will write $\Gamma_{x \to x'}(z)$ for the vector obtained by parallel transporting $z$ along any minimizing geodesic connecting $x$ and $x'$.

In this general context, we will be interested in solving the Riemannian optimization problem

$$\text{minimize}_{x \in \mathcal{M}} f(x) \qquad \text{(Opt)}$$

for some smooth *objective function* $f : \mathcal{M} \to \mathbb{R}$. We will also respectively write

$$v(x) := -\text{grad} f(x) \qquad \text{and} \qquad H(x) := \text{Hess}(f(x)) \qquad (1)$$

for the *negative (Riemannian) gradient* and the *(Riemannian) Hessian* of $f$ at $x$. Finally, in terms of solutions of (Opt), we will focus on the avoidance of *strict saddle points* of $f$, i.e., points $\hat{x} \in \mathcal{M}$ for which

$$v(\hat{x}) = 0 \quad \text{and} \quad \lambda_{\min}(H(\hat{x})) < 0 \qquad (2)$$

where $\lambda_{\min}$ denotes the minimum eigenvalue of the tensor in question. We will also say that a smooth compact component (in the sense of manifolds) of critical points of $f$ is a *strict saddle manifold* if there exist constants $c_\pm > 0$ such that all negative eigenvalues of $H(\hat{x})$, $\hat{x} \in \mathcal{S}$, are bounded from above by $-c_- < 0$, and any positive eigenvalues (if they exist) are bounded from below by $c_+ > 0$.

To differentiate the above from the Euclidean setting, when $\mathcal{M}$ is a real space equipped with the Euclidean metric, we will instead write $\nabla f$ and $\nabla^2 f$ for the (ordinary) gradient and Hessian matrix of $f$. In this case, as is customary, we will not distinguish between primal and dual vectors.

## 3 Core algorithmic framework

For generality, our avoidance analysis will be carried out in an abstract stochastic approximation framework which includes several popular Riemannian optimization algorithms – from ordinary Riemannian (stochastic) gradient descent, to its retraction-based variants, optimistic methods, etc. For concreteness, we start with the general template below, and we present a (nonexhaustive!) series of representative examples right after.

**3.1. The Riemannian Robbins–Monro template.** The *Riemannian Robbins–Monro* (RRM) framework that we will consider for solving (Opt) is an iterative family of methods which directly extends the seminal stochastic approximation scheme of Robbins & Monro [59] to a manifold setting by replacing vector addition with the Riemannian exponential. Roughly following [33], we will focus on the abstract update rule

$$X_{n+1} = \exp_{X_n}(\gamma_n \hat{v}_n) \qquad \text{(RRM)}$$

where

1. $X_n \in \mathcal{M}$ denotes the state of the algorithm at each iteration $n = 1, 2, \dots$
2. $\hat{v}_n \in \mathcal{T}_{X_n}\mathcal{M}$ is a surrogate for the (negative) gradient $v(X_n)$ of $f$ at $X_n$ (defined in detail below).
3. $\gamma_n > 0$ is the method's step-size (discussed in detail in Section 4).

In the above, the defining element of (RRM) is the sequence of "surrogate gradients" $\hat{v}_n$, $n = 1, 2, \dots$, so this will be our first object of interest. Formally, letting $\mathcal{F}_n$ denote the history of $X_n$ up to stage $n$ (inclusive), we will write

$$\hat{v}_n := v(X_n) + U_n + b_n \qquad (3)$$

where we have defined

$$U_n := \hat{v}_n - \mathbb{E}[\hat{v}_n \mid \mathcal{F}_n] \qquad \text{and} \qquad b_n := \mathbb{E}[\hat{v}_n \mid \mathcal{F}_n] - v(X_n), \qquad (4)$$

as the *random error* and the *offset* of $\hat{v}_n$ relative to $v(X_n)$ respectively. It will also be convenient to introduce the *total error* $W_n = \hat{v}_n - v(X_n) = U_n + b_n$, which captures both random and systematic fluctuations in $\hat{v}_n$, and which measures the total deviation of $\hat{v}_n$ from $v(X_n)$.

Two points are worth noting here: First, $\hat{v}_n$ is *not* adapted to $\mathcal{F}_n$, so $U_n$ is random relative to $\mathcal{F}_n$; on the other hand, $b_n$ is $\mathcal{F}_n$-measurable, so it is deterministic relative to $\mathcal{F}_n$. This brings us to the second important point regarding $\hat{v}_n$: given the systematic offset term $b_n$ in $\hat{v}_n$, the latter should *not be seen* as the output of a gradient oracle for $v(X_n)$. In particular, $b_n$ is intended to capture possible corrective terms, deviations from the exponential mapping, different algorithmic update structures (such as optimism), etc. We make this distinction precise below.

**3.2. Specific algorithms and examples.** In the series of examples that follow, we will assume that the optimizer can access $f$ via a *stochastic first-order oracle* (SFO) returning noisy gradients of $f$ at the evaluation point. Formally, following Nesterov [50], an SFO is a black-box mechanism which, when queried at $x \in \mathcal{M}$, returns a (negative) stochastic gradient of the form

$$V(x; \theta) = v(x) + \text{err}(x; \theta) \tag{SFO}$$

where the *seed* $\theta \in \Theta$ is a random variable taking values in some complete probability space $\Theta$, and $\text{err}(x; \theta)$ is an umbrella error term capturing all sources of uncertainty in the model.

The archetypal example of an SFO occurs when $f$ is itself a stochastic expectation of the form $f(x) = \mathbb{E}[F(x; \theta)]$ for some random function $F \colon \mathcal{M} \times \Theta \to \mathbb{R}$ – the so-called *stochastic optimization* framework. In this case, $V$ is typically given by $V(x; \theta) = -\text{grad}_x F(x; \theta)$, so, under standard assumptions for exchanging differentiation and expectation, we have $\mathbb{E}[V(x; \theta)] = v(x)$. Extrapolating from this basic framework, our only assumption for the moment will be that $\mathbb{E}[\text{err}(x; \theta)] = 0$; for a detailed discussion of the required assumptions for (SFO), see Section 4.

In practice, (SFO) will be queried repeatedly at a sequence of states $X_n$, $n = 1, 2, \ldots$, with a different random seed $\theta_n$ drawn i.i.d. from $\Theta$. In this manner, we obtain the following specific algorithms as special cases of (RRM):

**Algorithm 1** (Riemannian stochastic gradient descent)**.** Following Bonnabel [17], the *Riemannian stochastic gradient descent* (RSGD) algorithm queries (SFO) at $X_n$ and proceeds as

$$X_{n+1} = \exp_{X_n}(\gamma_n V(X_n; \theta_n)). \tag{RSGD}$$

As such, (RSGD) can be seen as an RRM scheme with $\hat{v}_n = V(X_n; \theta_n)$ or, equivalently, $U_n = \text{err}(X_n; \theta_n)$ and $b_n = 0$. ♦

A key factor limiting the applicability of (RSGD) is that the exponential map $\exp_{X_n}(\cdot)$ could be prohibitively expensive to compute in practice, even for relatively low-dimensional manifolds. On that account, a popular alternative to (RSGD) is to employ a *retraction map* [2, 19], that is, a smooth mapping $\mathcal{R} \colon \mathcal{TM} \to \mathcal{M}$ that agrees with the exponential map up to first order, namely

$$\mathcal{R}_x(0) = x \qquad \text{and} \qquad \frac{d}{dt}\Big|_{t=0} \mathcal{R}_x(tz) = z \quad \text{for all } (x, z) \in \mathcal{TM}. \tag{Rtr}$$

With this machinery in hand, we obtain the following retraction-based variant of (RSGD):

**Algorithm 2** (Retraction-based stochastic gradient descent)**.** By replacing the exponential map in (RSGD) with a retraction, we obtain the *retraction-based stochastic gradient descent* scheme

$$X_{n+1} = \mathcal{R}_{X_n}(\gamma_n V(X_n; \theta_n)). \tag{Rtr-SGD}$$

This algorithm does not seem immediately related to the RRM template – and, indeed, the whole point of introducing a retraction was to get rid of the exponential map in (RRM). The expressive power of (RRM) can be seen in the fact that, despite this apparent disconnect, (Rtr-SGD) can still be expressed as a special case of (RRM) in a fairly straightforward fashion.

To do so, define the "forward-backward" gradient mapping

$$\hat{v}_n \coloneqq \frac{1}{\gamma_n} \log_{X_n}(\mathcal{R}_{X_n}(\gamma_n V(X_n; \theta_n))) \tag{5}$$

with the proviso that the Riemannian logarithm in (5) is well-defined (we discuss the conditions under which this holds later in the paper). Under this definition, (Rtr-SGD) can be recast as a special case of (RRM) by running the latter with the surrogate gradient sequence $\hat{v}_n$ of Eq. (5). To streamline our presentation, we defer the discussion about the inherent error $W_n = \hat{v}_n - v(X_n)$ to Appendix A. ♦

As we mentioned before, retraction-based algorithms typically exhibit significantly lower per-iteration complexity compared to geodesic methods, resulting in their remarkable success in practical machine learning applications [2, 19]. In addition, as we show below, the use of a retraction mapping allows us to provide a unified perspective for several classical algorithms which, at first sight, might seem completely unrelated. An important example is provided by the (stochastic) *mirror descent* (MD) family of algorithms [49]:

**Algorithm 3** (Stochastic mirror descent). Let $\mathcal{M}$ be an open convex subset of $\mathbb{R}^M$ and let $h\colon \mathcal{M} \to \mathbb{R}$ be a $C^2$-smooth, strongly convex *Legendre function* on $\mathcal{M}$, that is, $\|\nabla h(x)\| \to \infty$ whenever $x \to \mathrm{bd}(\mathcal{M})$ [cf. 60, Chap. 26]. Then, the *stochastic mirror descent* (SMD) algorithm unfolds as

$$X_{n+1} = \mathcal{P}_{X_n}(\gamma_n V(X_n; \theta_n)) \qquad\qquad \text{(SMD)}$$

where $V(X_n; \theta_n)$ is the output of an SFO query for $\nabla f(X_n)$ at $X_n$, and $\mathcal{P}\colon \mathcal{M} \times \mathbb{R}^M \to \mathcal{M}$ is the so-called *prox-mapping* associated to $h$ [7–9, 31], viz.

$$\mathcal{P}_x(y) = \arg\max_{x' \in \mathcal{M}}\{\langle \nabla h(x) + y, x'\rangle - h(x')\} \qquad \text{for all } x \in \mathcal{M}, y \in \mathbb{R}^M. \qquad (6)$$

where $\langle \cdot, \cdot \rangle$ stands for the ordinary Euclidean inner product in $\mathbb{R}^M$.

Now, even though the notation in (SMD) is reminiscent of (Rtr-SGD), the definition (6) of $\mathcal{P}$ does not bear any resemblance to a geodesic exponential or a retraction – and, indeed, its origins are starkly different. However, as we show below, $\mathcal{P}$ can indeed be seen as a retraction relative to a specific Riemannian structure on $\mathcal{M}$, the *Hessian Riemannian* (HR) metric associated to $h$ [3, 16, 24, 45].

To make this precise, the first step is to note that the basic recursive structure $x^+ = \mathcal{P}_x(y)$ of (SMD) can be rewritten as

$$x^+ = \mathcal{P}_x(y) = \nabla h^*(\nabla h(x) + y) \qquad\qquad (7)$$

where $h^*(y) = \max_{x \in \mathcal{M}}\{\langle y, x\rangle - h(x)\}$ denotes the convex conjugate of $h$, and we have used Danskin's theorem [62] to write $\arg\max_{x \in \mathcal{M}}\{\langle y, x\rangle - h(x)\} = \nabla h^*(y)$. Then, if we endow $\mathcal{M}$ with the Hessian Riemannian metric $g(x) = \nabla^2 h(x)$, the Riemannian gradient of $f$ relative to $g$ becomes $\mathrm{grad} f(x) = [\nabla^2 h(x)]^{-1} \nabla f(x)$; more generally, given a cotangent (dual) vector $y$ to $\mathcal{M}$ at $x$, the corresponding tangent (primal) vector will be $z = g(x)^{-1}y = [\nabla^2 h(x)]^{-1}y$. In view of this, by inverting the relation $z = g(x)^{-1}y$, the abstract mirror descent recursion (7) can be rewritten as

$$x^+ = \mathcal{R}_x(z) \coloneqq \mathcal{P}_x(g(x)z). \qquad\qquad (8)$$

Now, to proceed, consider the curve

$$\gamma(t) = \mathcal{R}_x(tz) = \mathcal{P}_x(tg(x)z) = \nabla h^*(\nabla h(x) + tg(x)z), \qquad\qquad (9)$$

so, by definition, $\gamma(0) = x$. In addition, by a direct differentiation, we readily obtain

$$\dot{\gamma}(0) = \nabla^2 h^*(\nabla h(x))g(x)z = z \qquad\qquad (10)$$

where we used the standard identity $\nabla^2 h^*(\nabla h(x)) = [\nabla^2 h(x)]^{-1}$ [9, 61]. This shows that the map $\mathcal{R}_x(z) = \mathcal{P}_x(g(x)z)$ is, in fact, a *retraction*, so (SMD) is a special case of (Rtr-SGD) – and hence, of the general stochastic approximation template (RRM). ♦

*Remark.* Even though elements of the above ideas are implicit in previous works on mirror descent and Hessian Riemannian metrics [3, 5, 16, 58, 68], to the best of our knowledge, this is the first time that (SMD) is formalized as a retraction-based (Hessian) Riemannian scheme. ♦

**Algorithm 4** (Riemannian optimistic gradient). Moving forward, an important algorithm for solving online optimization problems and games is the so-called optimistic gradient method – originally pioneered by Popov [56] and subsequently popularized by Rakhlin & Sridharan [57]. In the Euclidean case, this method introduces an interim, "optimistic" correction to gradient dynamics and updates as

$$\begin{aligned} X_n^+ &= X_n + \gamma_n V(X_{n-1}^+; \theta_{n-1}) \\ X_{n+1} &= X_n + \gamma_n V(X_n^+; \theta_n) \end{aligned} \qquad\qquad \text{(OG)}$$

where, as usual, $V$ is an SFO for the (negative) gradient $\nabla f$ of $f$. This idea can then be directly transported to a manifold setting [33], leading to the *Riemannian optimistic gradient* method

$$\begin{aligned} X_n^+ &= \exp_{X_n}(\gamma_n V(X_{n-1}^+; \theta_{n-1})), \\ X_{n+1} &= \exp_{X_n}(\Gamma_{X_n^+ \to X_n}(\gamma_n V(X_n^+; \theta_n))). \end{aligned} \qquad\qquad \text{(ROG)}$$

Importantly, the recursion (ROG) may be seen as a special case of (RRM) by setting $\hat{v}_n = (1/\gamma_n) \cdot \Gamma_{X_n^+ \to X_n}(\gamma_n V(X_n^+; \theta_n))$ or, equivalently $U_n = \Gamma_{X_n^+ \to X_n}(\mathrm{err}(X_n^+; \theta_n))$ and $b_n = \Gamma_{X_n^+ \to X_n}(v(X_n^+)) - v(X_n)$. We defer the details of this calculation to the Appendix A. ♦

**Algorithm 5** (Natural gradient descent)**.** Our last example concerns the influential *natural gradient descent* (NGD) method of Amari [4], a stochastic optimization scheme for Euclidean spaces, but adapted to the local geometry defined by a strictly convex function $h$. Specifically, NGD queries an SFO and proceeds as

$$X_{n+1} = X_n - \gamma_n(\operatorname{grad} f(X_n) + \operatorname{err}(X_n; \theta_n)) \qquad \text{(NGD)}$$

where $\operatorname{grad} f(x) \coloneqq [\nabla^2 h(x)]^{-1} \nabla f(x)$ denotes the Riemannian gradient of $f$ relative to Hessian Riemannian metric $g(x) = \nabla h^2(x)$ on $\mathbb{R}^M$. It is well known that (NGD) can be seen as a retraction-based Riemannian scheme [17], and may thus be integrated directly within the framework of (RRM); we defer the details to Appendix A. Importantly, (NGD) also includes the celebrated *natural policy gradient* [32] which plays an important role in reinforcement learning. ♦

The above examples have been chosen to illustrate a range of different update mechanisms that can be integrated within the general algorithmic template provided by (RRM). Of course, it is not possible to be exhaustive but, for illustration purposes, we provide some more examples in Appendix A.

## 4 Analysis and results

We are now in a position to state and discuss our main result concerning the avoidance of saddle points under (RRM). For concreteness, we begin by discussing the technical assumptions that we will need in Section 4.1; subsequently, we proceed with the formal statement of our result and some direct applications thereof in Section 4.2.

**4.1. Technical assumptions.** Our technical assumptions concern the four main ingredients of (RRM), namely (*i*) the regularity of $f$; (*ii*) the method's step-size sequence $\gamma_n$; (*iii*) the statistics of the surrogate gradients $\hat{v}_n$ entering (RRM); and (*iv*) the ambient manifold $\mathcal{M}$. Specifically, we will require the following:

**Assumption 1** (Regularity of $f$)**.** The function $f$ is $C^2$ and $v = -\operatorname{grad} f$ is (geodesically) *L-Lipschitz*, i.e., for all $x, x' \in \mathcal{M}$, we have

$$\|\Gamma_{x \to x'}(v(x)) - v(x')\|_{x'} = \|v(x) - \Gamma_{x' \to x}(v(x'))\|_x \le L \operatorname{dist}(x, x'). \qquad (11)$$

**Assumption 2** (Step-size schedule)**.** The step-size sequence $\gamma_n$ of (RRM) satisfies

$$\sum_{n=1}^{\infty} \gamma_n = \infty \quad \text{and} \quad \sum_{n=1}^{\infty} \lambda^{1/\gamma_n} < \infty \quad \text{for all } \lambda \in (0, 1). \qquad (12)$$

**Assumption 3** (Surrogate gradients)**.** The offset and random error components of $\hat{v}_n$ satisfy

$$\|b_n\|_{x_n} \le C\gamma_n, \qquad \|U_n\|_{x_n} \le \sigma, \qquad \mathbb{E}[[\langle U_n, z \rangle_{x_n}]_+ \,|\, \mathcal{F}_n] \ge \zeta \qquad (13)$$

for suitable constants $C, \sigma, \zeta > 0$ and for all $z \in \mathcal{T}_{X_n}\mathcal{M}$, $\|z\|_{X_n} = 1$ (in the above, all conditions are to be interpreted in the almost sure sense and $[t]_+ = \max\{0, z\}$ denotes the positive part of $t$).

**Assumption 4** (Injectivity radius)**.** The injectivity radius of $\mathcal{M}$ is bounded from below by $\varrho > 0$.

Before proceeding, we discuss the implications and range of validity of each of the above assumptions. Since Assumption 1 is standard, we focus on the remaining three below:

**On Assumption 2.** The step-size conditions typically encountered in the analysis of Robbins–Monro schemes is the $L^2 - L^1$ ("*square-summabe-but-not-summable*") condition $\sum_n \gamma_n = \infty$, $\sum_n \gamma_n^2 < \infty$, cf. [10, 13, 14, 17, 39, 59] and references therein. This puts a hard threshold on the range of allowed step-size schedules at $\Omega(1/n^{1/2})$: any step-size that decays at least as slow as $1/n^{1/2}$ cannot be used under the $L^2 - L^1$ assumption. By contrast, the step-size condition (12) is considerably more lax and can tolerate near-constant step-sizes of the form $\gamma_n \propto 1/(\log n)^{1+\varepsilon}$ for some $\varepsilon > 0$. This is enough to cover all dereasing step-size policies used in practice. [We also recall here that, in stochastic non-convex settings, trajectory convergence cannot be guaranteed in general without a vanishing step-size, cf. [10, 18, 40] and references therein.]

**On Assumption 3.** Three remarks are in order for the noise and offset requirements (13). First, we should note that the condition $b_n = \mathcal{O}(\gamma_n)$ is, a priori, *implicit*, because it depends on the statistics of the feedback sequence $\hat{v}_n$, and these may be difficult to estimate in general. However, in most practical applications, this quantity is under the *explicit* control of the optimizer: in particular, as we show later in this section, this requirement is satisfied by all the specific algorithms of Section 3.2.

Likewise, the bounded noise requirement is satisfied in many practical cases of interest. For example, when the problem's objective function admits a finite-sum decomposition of the form $f(x) = \sum_{i=1}^{N} f_i(x)$ for an ensemble of empirical instances $f_i$, $i = 1, \ldots, N$ (the standard framework for applications to data science and machine learning), $U_n$ is typically generated by sampling a minibatch of $f$, which in turn results in an error term of the form $U_n = q(X_n)$ where $q(x) : \mathcal{M} \to \mathcal{T}_x\mathcal{M}$ is bounded on all compact subsets of $\mathcal{M}$. Therefore, $\|U_n\|_{x_n} \le \|q(X_n)\|_{x_n} < \sigma$ for some constant $\sigma$ for any convergent algorithm $\{X_n\}_n$.

Finally, the "uniform excitability" condition $\mathbb{E}[[\langle U_n, z \rangle_{x_n}]_+ \mid \mathcal{F}_n] \ge \zeta$ is also standard in the avoidance literature [10, 54], and it is substantially weaker than the *isotropic* condition, which, roughly speaking, requires the noise to have the same $L^2$ magnitude along all directions in space [26, 30, 54]. Instead, (13) only posits that the noise $U_n$ has a *non-zero* component along each direction, and imposes no other restrictions on the statistical profile of the noise.

**On Assumption 4.** For our last assumption, recall first that the injectivity radius of $\mathcal{M}$ at a point $x \in \mathcal{M}$ is the largest radius for which $\exp_x$ is a diffeomorphism onto its image; the injectivity radius of $\mathcal{M}$ is then taken to be the infimum over all such radii [43]. In this regard, Assumption 4 simply serves to ensure that the exponential map is invertible at consecutive iterates of (RRM) so no local topological complications can arise. This assumption is automatically satisfied in closed manifolds (independent of curvature), as well as in non-positively curved manifolds – such as Cartan-Hadamard spaces and the like [43, 44]. This assumption (and its variants) is also standard in the literature, cf. [17, 34, 66] and references therein.

**4.2. Avoidance of saddle points.** We are now in a position to state our main avoidance result:

**Theorem 1.** *Let $X_n$, $n = 1, 2, \ldots$, be the sequence of states generated by* (RRM)*, and let $\mathcal{S}$ be a strict saddle manifold of $f$. Then, under Assumptions 1–4, we have*

$$\mathbb{P}(\mathrm{dist}(\mathcal{S}, X_n) \to 0 \text{ as } n \to \infty) = 0 \tag{14}$$

*where $\mathrm{dist}(\mathcal{S}, X_n) = \inf_{x \in \mathcal{S}} \mathrm{dist}(x, X_n)$ denotes the (Riemannian) distance of $X_n$ from $\mathcal{S}$.*

Before discussing the proof of Theorem 1, it is worthwhile to compare our work with its closest antecedents. First, in regard to the general avoidance theory in Euclidean spaces [10, 12], the statement is similar in scope (avoidance of unstable manifolds with probability 1), but the techniques and challenges involved are very different. The reason for this is simple: the additive, vector space structure of $\mathbb{R}^m$ is ingrained at every step of the way in the Euclidean analysis of [10], and adapting the various constructions to a manifold setting can be a complicated affair. For an illustration of the technical difficulties involved, see the recent stochastic approximation analysis of [33].

By contrast, the recent results of [22, 66] paint a complementary picture: they concern Riemannian problems but, at their core, they are deterministic results. More precisely, the noise in [22, 66] is actually *injected* in an otherwise deterministic gradient scheme to facilitate the escape from flat regions in the vicinity of a saddle point; other than that, the magnitude of the noise must be proportional to the solver's desired accuracy, and hence is typically extremely small. As a result, the analysis of [22, 66] cannot be extended to bona fide stochastic schemes – like (RSGD) – which also explains why these results involve a constant step-size (as opposed to a decreasing step-size schedule, which is required to guarantee trajectory convergence in settings with persistent noise). In this regard, Theorem 1 simultaneously complements the stochastic analysis of [10, 12] to Riemannian problems, and the Riemannian analysis of [22, 66] to a stochastic setting.

**Proof outline.** To facilitate the reading of our proof, we provide below a detailed outline of the main steps and techniques involved therein, deferring the full proof to Appendix B. We begin with a high-level description of our proof strategy and then encode the main arguments in a series of steps right after.

For the purposes of illustration, suppose that $\mathcal{M}$ is a subset of $\mathbb{R}^m$. Then, given a tangent vector $z \in \mathcal{T}_x\mathcal{M}$, we define the *geodesic offset* (see Fig. 1) from $x$ along $z$ as

$$\Delta(x; z) = \exp_x(z) - x - z \tag{15}$$

i.e., as the difference between the geodesic emanating from $x$ along $z$ and its first-order approximation relative to $x$ in the ambient space $\mathbb{R}^m$ (with all differences expressed in the ordinary vector space structure of $\mathbb{R}^m$). The offset $\Delta(x; z)$ is readily checked to be second-order in $z$ so, while the curve $x + tz$ does not in general induce a retraction on the target manifold $\mathcal{M}$ (in particular, the point $x + tz$ may not even *belong* to $\mathcal{M}$), the converse *is* true: the map $\exp_x(z)$ is always a retraction on the ambient, Euclidean space $\mathbb{R}^m$. In this way, the basic iteration (RRM) can be expressed as

$$X_{n+1} = \exp_{X_n}(\gamma_n \hat{v}_n) = X_n + \gamma_n \hat{v}_n + \Delta(X_n; \gamma_n \hat{v}_n) \tag{16}$$

leading to the fundamental question below:

> *What is the maximum offset $\epsilon_n \coloneqq \Delta(X_n; \gamma_n\hat{v}_n)$ that can be tolerated by a Euclidean stochastic approximation algorithm to avoid saddle points?*

A key technical step in our work is to develop the means to control the offset term $\epsilon_n$ under (RRM) under a sufficiently broad class of assumptions that includes Algorithms 1–5. Crucially, this step is made possible thanks to the very recent – and technical – stochastic approximation work of [33]. To help the reader navigate our proof strategy, we outline the main steps below, focusing for simplicity on the case of a single saddle point.

**Step 1: From discrete to continuous time (and back).**   Let $\hat{x}$ be a strict saddle point of $f$. By the stable manifold theorem [63], the set of all initializations such that the Riemannian gradient flow

$$\dot{x}(t) = -\mathrm{grad} f(x(t)) \tag{RGF}$$

converges to $\hat{x}$ is of measure 0. Then, assuming for the moment that the geodesic offset error $\epsilon_n = \Delta(X_n; \gamma_n\hat{v}_n)$ in (16) is sufficiently small, the iterates of (RRM) can be seen as a noisy, approximate Euler discretization of (RGF); as such, it is reasonable to expect that the induced trajectories of (RRM) will never converge to $\hat{x}$.

To make this intuition precise, our first step will be to show that the iterates of (RRM) comprise an *asymptotic pseudotrajectory* of (RGF) in the sense of Benaïm [10], i.e., they asymptotically track the orbits of (RGF) with arbitrary precision over windows of arbitrary length. To formalize this, define the "effective time" variable $\tau_n = \sum_{k=1}^{n-1} \gamma_k$ and the associated *geodesic interpolation* $X(t)$ of $X_n$ as

$$X(t) = \exp_{X_n}((t - \tau_n)\hat{v}_n) \quad \text{for all } t \in [\tau_n, \tau_{n+1}), n \geq 1 \tag{GI}$$

so, by construction, (*a*) $X(\tau_n) = X_n$ for all $n$; and (*b*) each segment of $X(t)$ is a geodesic. Then, letting $\Phi \colon \mathbb{R}_+ \times \mathcal{M} \to \mathcal{M}$ denote the *flow* of (RGF) – i.e., $\Phi_h(x)$ is simply the position at time $h \geq 0$ of the solution orbit of (RGF) that starts at $x \in \mathcal{M}$ – we will say that $X(t)$ is an APT of (RGF) if, for all $T > 0$, we have

$$\lim_{t\to\infty} \sup_{0 \leq h \leq T} \mathrm{dist}(X(t+h), \Phi_h(X(t))) = 0. \tag{APT}$$

This requirement is non-trivial, and our first technical result is to guarantee precisely this:

**Theorem 2.** *Suppose that Assumptions 2–4 hold. Then, with probability 1, the geodesic interpolation $X(t)$ of the sequence of iterates $X_n$, $n = 1, 2, \ldots$, generated by (RRM) is an APT of (RGF).*

A version of Theorem 2 was very recently derived by [33] under a different set of assumptions: On the one hand, [33] imposes a much more restrictive step-size schedule for $\gamma_n$ (square summability) but, on the other hand, it only posits that the noise increments $U_n$ are bounded in $L^2$ (as opposed to $L^\infty$ in our case). Our proof relies on the same construction of the Picard iteration map as [33], but otherwise diverges significantly in the probabilistic analysis required to establish (APT).

**Step 2: From Riemannian to Euclidean schemes (and back).**   Albeit crucial, the APT property is decidedly not enough to guarantee avoidance: after all, the constant orbit $X(t) = \hat{x}$ for all $t \geq 0$ is trivially an APT of (RGF) but, of course, it does not avoid $\hat{x}$. To proceed, we will need to exploit the precise update structure of (RRM) in conjunction with the stable manifold theorem applied to (RGF).

In the Euclidean case, this is achieved by means of an intricate Lyapunov function argument, originally due to [11]. Our second step is to devise a new geometric argument to reduce the analysis from an arbitrary *intrinsic* manifold to an isometrically embedded submanifold of $\mathbb{R}^m$. This step is carried out by a combination of the celebrated Nash embedding theorem and a (smooth) Tietze extension argument to rewrite (RRM) as a "corrected" Robbins–Monro scheme on $\mathbb{R}^m$ that actually evolves on $\mathcal{M}$. This construction also requires a "perturbation analysis" to ensure that certain subtle topological issues do not arise when we invoke the stable manifold theorem; we present the details in Appendix B.

**Step 3: Controlling the geodesic offset.** As we briefly described in the beginning of the proof overview, this Euclidean reframing of (RRM) introduces an intrinsic offset error $\epsilon_n = \Delta(X_n; \gamma_n \hat{v}_n)$, which is difficult to analyze in detail (the offset incurred by a retraction on $\mathcal{M}$ is of similar order, so the exponential-retraction distinction is not important at this stage). Our crucial observation here is that, under our blanket assumptions, $\epsilon_n$ is small relative to $\gamma_n$ and, in particular, $\epsilon_n = \mathcal{O}(\gamma_n^2)$. Thanks to this bound, we are able to leverage a series of stochastic bounds – originally developed by Pemantle [54] – to show that the probability that these terms will have an adverse effect on exiting the center manifold of $\hat{x}$ is zero (this is also where Assumption 3 comes in). We formalize this in Appendix B; Theorem 1 then follows by putting everything together.

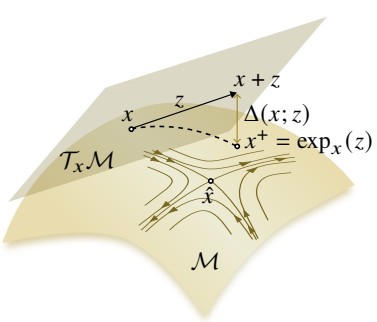

**Figure 1:** The geodesic offset $\Delta(x; z)$.

*Remark.* A concept similar to our geodesic offset $\Delta(x; z)$ has been explored in the *reverse direction* by the very recent works [15, 23] whose goal was to study of avoidance of *Euclidean* subgradient methods as an inexact Riemannian gradient scheme. They further show that this inexact Riemannian gradient descent can avoid saddle points if uniform noise is injected. While their core idea bears some similarity to ours, it remains unclear how to apply the analysis in [15, 23] to handle general RRM schemes, such as retraction-based and natural policy gradient methods.

**4.3. Applications.** As an illustration of the generality of Theorem 1, we now instantiate it to the range of specific algorithms discussed in Section 3.2. Since all these algorithms are run with gradient input generated by (SFO), applying Theorem 1 would require mapping the requirements of Assumption 3 to the primitives of (SFO). A convenient way to achieve this is by means of the proposition below:

**Proposition 1.** *Suppose that Algorithms 1–5 are run with a gradient oracle $V(x; \theta) = v(x) + \mathrm{err}(x; \theta)$ such that*

$$\|\mathrm{err}(x; \theta)\|_x \leq \sigma(x) \quad and \quad \mathbb{E}[[\langle \mathrm{err}(x; \theta), z \rangle_x]_+] \geq \zeta(x) \tag{17}$$

*for all $z \in \mathcal{T}_x \mathcal{M}$, $\|z\|_x = 1$, and for suitable functions $\zeta, \sigma \colon \mathcal{M} \to \mathbb{R}_+$ with $\sigma$ bounded on bounded subsets of $\mathcal{M}$ and $\inf_x \zeta(x) > 0$. Then, under Assumptions 2 and 4, the conclusion of Theorem 1 holds, that is, Algorithms 1–5 avoid strict saddle manifolds of $f$.*

The proof of Proposition 1 is deferred to Appendix B; we only note here that its proof mainly hinges on verifying the bias requirement $\|b_n\|_{X_n} = \mathcal{O}(\gamma_n)$ of (13) by means of (*i*) the boundedness of the error function $\sigma(x)$ on bounded subsets of $\mathcal{M}$; and (*ii*) controlling the maximal deviation between a retraction and the exponential map for input vectors bounded by $\varrho$.

# 5   Numerical Illustrations

In this section, we aim to demonstrate the practical applicability of the theoretical framework proposed in our paper by providing numerical illustrations. To do so, we utilize a 2-dimensional torus as the optimization landscape, where the complexity and multi-modal nature of the objective function can be easily visualized in Fig. 2.

Our objective function features three saddle points (in black) and one global minimizer (in red). We subject two RRM schemes, i.e., (RSGD) and (RSEG), to initialization in proximity to these saddle points. This strategic choice rigorously tests their ability to navigate and converge to the global optimum. In line with our theoretical predictions, both RRM methods avoid the saddle points and

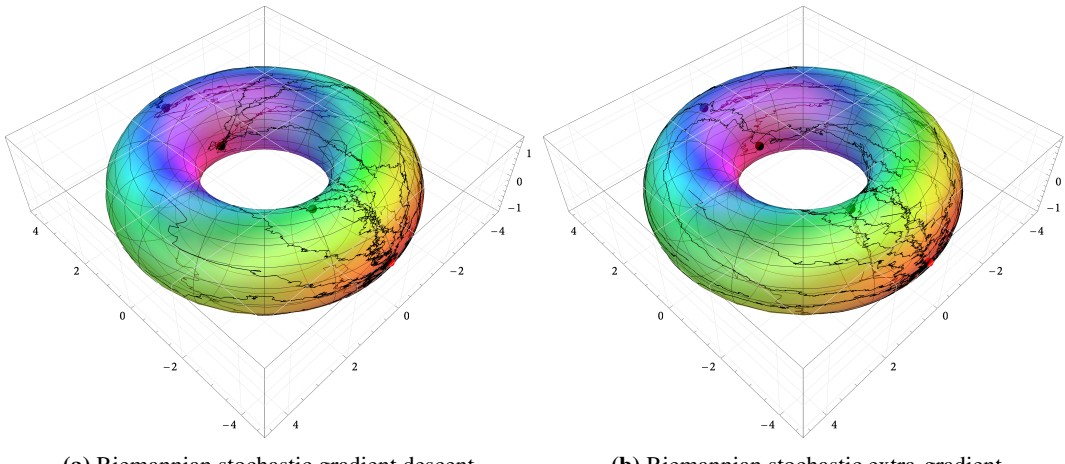

**(a)** Riemannian stochastic gradient descent      **(b)** Riemannian stochastic extra-gradient

**Figure 2:** Stochastic RRM schemes on the torus.

eventually reach the desirable global minimum. This empirical confirmation reinforces the central message of our paper: stochastic Riemannian schemes only converge to local minimizers.

# 6    Conclusions and future work

In this paper, we addressed the question of when Riemannian stochastic algorithms can effectively evade saddle points, focusing on the broad category of Riemannian Robbins–Monro schemes. We introduced a novel framework for analyzing the avoidance of Riemannian saddle points within the RRM framework, which encompasses many commonly used Riemannian stochastic algorithms, including retraction-based algorithms. Our framework builds upon the notion of strict saddle points and provides a set of easily verifiable conditions that guarantee the avoidance of such traps.

Our work paves the way for several promising research directions in learning with Riemannian methods. One intriguing avenue for exploration is the investigation of whether Riemannian *zeroth-order* methods, such as the Riemannian extension of the work by Kiefer & Wolfowitz [36], can effectively evade strict saddle points. We believe that combining the insights from the asymptotic pseudotrajectory theory with Euclidean analysis can shed light on this question and provide valuable insights into the behavior of these methods in the Riemannian setting.

Furthermore, an interesting direction for future research is the extension of the avoidance of *unstable limit cycles* in Euclidean min-max optimization, as studied by Hsieh et al. [29], to the realm of Riemannian games. Investigating the avoidance of unstable limit cycles in this context has the potential to uncover novel phenomena specific to the manifold settings, leading to a deeper understanding on the intricate dynamics and strategies involved in Riemannian games.

## Acknowledgments and Disclosure of Funding

This work has been partially supported by the European Research Council (ERC) under the European Union's Horizon 2020 research and innovation program grant agreement No 815943, by project MIS 5154714 of the National Recovery and Resilience Plan Greece 2.0 funded by the European Union under the NextGenerationEU Program, and by the French National Research Agency (ANR) in the framework of the PEPR IA FOUNDRY project (ANR-23-PEIA-0003), the "Investissements d'avenir" program (ANR-15-IDEX-02), the LabEx PERSYVAL (ANR-11-LABX-0025-01), and MIAI@Grenoble Alpes (ANR-19-P3IA-0003). PM is also with the Archimedes Research Unit – Athena RC – University of Athens. YPH acknowledges funding through an ETH Foundations of Data Science (ETH-FDS) postdoctoral fellowship.

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

# A  Further examples of RRM schemes

In this section, we provide two additional algorithmic examples supplementing the range of Algorithms 1–5 to illustrate the applicability of our RRM template.

**Algorithm 6** (Riemannian proximal point methods)**.**  The (deterministic) *Riemannian proximal point method* (RPPM) [25] is an implicit ("backward") update rule of the form

$$\log_{X_{n+1}}(X_n) = -\gamma_n v(X_{n+1}). \tag{RPPM}$$

The RRM representation of (RPPM) is then obtained by taking $b_n = \Gamma_{X_{n+1} \to X_n}(v(X_{n+1})) - v(X_n)$ and $U_n = 0$ in the decomposition (4) of the error term $W_n$ of (RRM). If, in additional, the true gradient $v(X_{n+1})$ is replaced by an oracle $V(X_{n+1}; \theta_{n+1})$, then (RPPM) becomes the stochastic version of RPPM by setting

$$b_n = \mathbb{E}[\Gamma_{X_{n+1} \to X_n}(V(X_{n+1}; \theta_{n+1})) - v(X_n) \mid \mathcal{F}_n]$$

and

$$U_n = \Gamma_{X_{n+1} \to X_n}(V(X_{n+1}; \theta_{n+1})) - v(X_n) - b_n.$$

For a detailed discussion, see [25] and references therein.  ◆

**Algorithm 7** (Riemannian stochastic extra-gradient)**.**  Inspired by the original work of Korpelevich [38], the *Riemannian stochastic extra-gradient* (RSEG) method [51, 67] proceeds as

$$\begin{aligned}
X_n^+ &= \exp_{X_n}(\gamma_n V(X_n; \theta_n)), \\
X_{n+1} &= \exp_{X_n}(\Gamma_{X_n^+ \to X_n}(\gamma_n V(X_n^+; \theta_n^+)))
\end{aligned} \tag{RSEG}$$

where $\theta_n$ and $\theta_n^+$ are independent seeds for (SFO). Thus, to cast (RSEG) in the RRM framework, it suffices to take $U_n = \Gamma_{X_n^+ \to X_n}(\text{err}(X_n^+; \theta_n^+))$ and $b_n = \Gamma_{X_n^+ \to X_n}(v(X_n^+)) - v(X_n)$.  ◆

Under the assumptions in Proposition 1, one can show that Algorithms 6–7 also avoid strict saddle points of $f$; we provide the relevant details in Appendix B.3.

# B  Missing Proofs of Section 4

**B.1. Proof of Theorem 2.**  We begin by proving Theorem 2, which will play a crucial role in our proof of Theorem 1. For the reader's convenience, we restate the result below:

**Theorem 2.**  *Suppose that Assumptions 2–4 hold. Then, with probability* 1*, the geodesic interpolation $X(t)$ of the sequence of iterates $X_n$, $n = 1, 2, \ldots$, generated by* (RRM) *is an APT of* (RGF).

*Proof.*  To begin, let $\{e_i(n)\}_{i=1}^d$ be an arbitrary sequence of orthonormal bases for $\mathcal{T}_{X_n}\mathcal{M}$, and let $U_n^{\parallel}$ be the (Euclidean) noise vector composed of components of the noise $U_n$ in the basis $\{e_i(n)\}_{i=1\ldots d}$, viz.

$$U_{i,n}^{\parallel} := \langle U_n, e_i(n) \rangle_{X_n}. \tag{B.1}$$

It is then easy to see that $\mathbb{E}[U_n^{\parallel} | \mathcal{F}_n] = 0$, and, moreover

$$\left\| U_n^{\parallel} \right\| = \| U_n \|_{X_n} \leq \sigma \tag{B.2}$$

by Assumption 3. Then, following Benaïm [10], consider the "continuous-to-discrete" counter

$$M(t) = \sup\{n \geq 1 : t \geq \tau_n\} \tag{B.3}$$

which measures the number of iterations required for the effective time $\tau_n = \sum_{k=1}^{n-1} \gamma_k$ to reach a given timestamp $t \geq 0$. We further denote the piecewise-constant interpolation of the noise sequence as

$$\bar{U}^{\parallel}(t) = U_n^{\parallel} \quad \text{for all } t \in [\tau_n, \tau_{n+1}), n \geq 1 \tag{B.4}$$

and we let

$$\Delta(t; T) := \sup_{0 \leq h \leq T} \left\| \int_t^{t+h} \bar{U}^{\parallel}(s) \, ds \right\|. \tag{B.5}$$

Moving forward, since $X_n \to \mathcal{S}$ by assumption, we will also have

$$\sup_n \text{dist}(X_n, x) =: R < \infty \quad \text{for all } x \in \mathcal{S} \tag{B.6}$$

for some (possibly random) $R \geq 0$. Moreover, since $f$ is assumed to be $C^2$, (B.6) implies that

$$\sup_n \|v(X_n)\|_{X_n} =: G < \infty \tag{B.7}$$

for some (possibly random) non-negative constant $G \geq 0$. Moreover, since the manifold $\mathcal{M}$ is assumed to be smooth, the sectional curvatures at each $X_n$, $n = 1, 2, \ldots$, must be likewise bounded by some constant $K_{\max}$. Then, by the analysis of [34, Eq. 53], there exists a constant $C \equiv C_{L,G,K_{\max},R}$ depending only on $L, G, K_{\max}$ and $R$ such that

$$\sup_{0 \leq h \leq T} \mathrm{dist}(X(t+h), \Phi_h(X(t))) \leq C_{L,G,K_{\max},R} \cdot \left[ \sup_{n \geq M(t)} (\|b_n\|_{X_n} + \gamma_n) + \Delta(t-1; T+1) \right] \tag{B.8}$$

By Assumption 3, we have $\|b_n\|_{X_n} = \mathcal{O}(\gamma_n)$. Since $\gamma_n \to 0$, it suffices to show that $\Delta(t; T) \to 0$ with probability 1 under Assumptions 2 and 3. This is equivalent to showing that, for any $\varepsilon > 0$, we have

$$\lim_{t \to \infty} \Delta(t; T) \leq \varepsilon \quad \text{with probability 1.} \tag{B.9}$$

To this end, let $n = M(t)$ and recall that, by (B.2), we have

$$\mathbb{E}\left[ \exp\left( \langle w, \bar{U}_n^{\shortparallel} \rangle \right) \big| \mathcal{F}_n \right] \leq \exp\left( \frac{\sigma^2}{2} \|w\|^2 \right) \tag{B.10}$$

for all $w \in \mathbb{R}^d$ where $d$ is the dimension of $\mathcal{M}$. Therefore, for each $w \in \mathbb{R}^d$, the sequence $Y_n(w)$ defined by

$$Y_n(w) := \exp\left( \sum_{k=1}^n \langle w, \gamma_k \bar{U}_k^{\shortparallel} \rangle - \frac{\sigma^2 \|w\|^2}{2} \sum_{k=1}^n \gamma_k^2 \right) \tag{B.11}$$

is a supermartingale. Since $Y_n$ is a supermartingale, we have

$$\mathbb{P}\left( \sup_{n < k \leq M(\tau_n + T)} \sum_{i=n}^{k-1} \langle w, \gamma_i \bar{U}_i^{\shortparallel} \rangle \geq \delta \right)$$

$$\leq \mathbb{P}\left( \sup_{n < k \leq M(\tau_n + T)} Y_k(w) \geq Y_n(w) \exp\left( \delta - \frac{\sigma^2 \|w\|^2}{2} \sum_{i=n}^{M(\tau_n + T)-1} \gamma_i^2 \right) \right)$$

$$\leq \exp\left( \frac{\sigma^2 \|w\|^2}{2} \sum_{i=n}^{M(\tau_n + T)-1} \gamma_i^2 - \delta \right) \tag{B.12}$$

for any $\delta > 0$. Now, let $e_i$ be the $i$-th basis vector of $\mathbb{R}^d$. Then, by (B.12), we have

$$\mathbb{P}\left( \sup_{n < k \leq M(\tau_n + T)} \sum_{i=n}^{k-1} \langle \pm d e_i, \gamma_k \bar{U}_k^{\shortparallel} \rangle \geq \varepsilon \right) = \mathbb{P}\left( \sup_{n < k \leq M(\tau_n + T)} \sum_{i=n}^{k-1} \langle \pm \varepsilon^{-1} \delta d e_i, \gamma_k \bar{U}_k^{\shortparallel} \rangle \geq \delta \right)$$

$$\leq \exp\left( \frac{\sigma^2 \delta^2 d^2}{2\varepsilon^2} \sum_{i=n}^{M(\tau_n + T)-1} \gamma_i^2 - \delta \right). \tag{B.13}$$

Optimizing (B.13) over $\delta$, we get

$$\mathbb{P}\left( \sup_{n < k \leq M(\tau_n + T)} \sum_{i=n}^{k-1} \langle \pm d e_i, \gamma_k \bar{U}_k^{\shortparallel} \rangle \geq \varepsilon \right) \leq \exp\left( - \frac{\varepsilon^2}{2\sigma^2 d^2 \sum_{i=n}^{M(\tau_n + T)-1} \gamma_i^2} \right). \tag{B.14}$$

Since $\gamma_n \to 0$, with loss of generality we may assume that $\gamma_n \leq 1$, and hence

$$\mathbb{P}\left( \sup_{n < k \leq M(\tau_n + T)} \sum_{i=n}^{k-1} \langle \pm d e_i, \gamma_k \bar{U}_k^{\shortparallel} \rangle \geq \varepsilon \right) \leq \exp\left( - \frac{\varepsilon^2}{2\sigma^2 d^2 \sum_{i=n}^{M(\tau_n + T)-1} \gamma_i} \right)$$

$$\leq \exp\left( - \frac{\varepsilon^2}{2\sigma^2 d^2 \int_t^{t+T} \bar{\gamma}(t)} \right) \tag{B.15}$$

where, analogously to (B.4), we have defined the piece-wise constant interpolated step-size sequence

$$\bar{\gamma}(t) = \gamma_n \quad \text{for all } t \in [\tau_n, \tau_{n+1}), n \geq 1. \tag{B.16}$$

Since

$$\left\| \sum_{i=n}^{k-1} \gamma_i \bar{U}_i^{\shortparallel} \right\| \geq \varepsilon \quad \Rightarrow \quad \exists i \text{ such that } \sum_{i=n}^{k-1} \langle \pm de_i, \gamma_i \bar{U}_i^{\shortparallel} \rangle \geq \varepsilon, \tag{B.17}$$

by the union bound, we have

$$\mathbb{P}(\Delta(t, T) \geq \varepsilon) \leq 2d \cdot \exp\left( -\frac{\varepsilon^2}{2\sigma^2 d^2 \int_t^{t+T} \bar{\gamma}(t)} \right) \leq 2d \cdot \exp\left( -\frac{\varepsilon^2}{2\sigma^2 d^2 T \bar{\gamma}(s)} \right) \tag{B.18}$$

for some $t \leq s \leq t + T$. Therefore, by setting $\lambda := \exp\left( -\frac{\varepsilon^2}{2\sigma^2 d^2 T} \right) < 1$, we have

$$\sum_k \mathbb{P}(\Delta(kT, T) \geq \varepsilon) \leq 2d \sum_k \lambda^{1/\gamma_k} < \infty \tag{B.19}$$

by Assumption 2. The Borel-Cantelli Lemma then implies that the following event happens almost surely:

$$\lim_{k \to \infty} \Delta(kT; T) \leq \varepsilon. \tag{B.20}$$

The proof is finished by noting that, for $kT \leq t < (k+1)T$,

$$\Delta(t; T) \leq 2\Delta(kT; T) + \Delta(kT + T; T) \tag{B.21}$$

by triangle inequality. ∎

**B.2. Proof of Theorem 1.** We are now in a position to present our proof of Theorem 1, which we restate below for convenience:

**Theorem 1.** *Let $X_n$, $n = 1, 2, \ldots$, be the sequence of states generated by (RRM), and let $S$ be a strict saddle manifold of $f$. Then, under Assumptions 1–4, we have*

$$\mathbb{P}(\text{dist}(S, X_n) \to 0 \text{ as } n \to \infty) = 0 \tag{14}$$

*where $\text{dist}(S, X_n) = \inf_{x \in S} \text{dist}(x, X_n)$ denotes the (Riemannian) distance of $X_n$ from $S$.*

*Proof.* Assume that $X_n \to S$. We will show that this event has zero probability in a series of steps which we outline below.

**Step 1: Isometrically embedded Robbins–Monro iterates.** Since $\mathcal{M}$ is assumed to be smooth, the second Nash embedding theorem [37] implies there exists a smooth and *isometric* embedding $\iota : \mathcal{M} \to \mathbb{R}^M$ such that, for all $x \in \mathcal{M}$ and all $z, w \in \mathcal{T}_x \mathcal{M}$, we have

$$\langle z, w \rangle_x = \langle \mathrm{D}\, \iota_x(z), \mathrm{D}\, \iota_x(w) \rangle. \tag{B.22}$$

Since $\iota$ is an embedding, it is surjective. Since it is isometric, it preserves distance and hence must be one-to-one. Therefore, $\iota$ is an diffeomorphism since it is also smooth. We can therefore define the *pushforward* of the vector field $v$ on $\mathcal{M}$ to a vector field on the image $\mathcal{M}^E \subset \mathbb{R}^M$ in the usual way as

$$v_0^E(x^E) := \mathrm{D}\, \iota_x v(x) \qquad \text{for all } x^E = \iota(x) \in \mathbb{R}^M. \tag{B.23}$$

We also set $S^E := \iota(S)$.

By the Tietze extension theorem and the smooth manifold extension lemma [44], $v_0^E(x^E)$ can be extended to a Lipschitz continuous vector field on all of $\mathbb{R}^M$, which we still denote by $v_0^E(x^E)$. To avoid trivialities, we will also need to ensure that $v_0^E(x^E)$ is not 0 in a neighborhood of $S^E$: If this is the case, then we set our target field $v^E(x^E) := v_0^E(x^E)$; otherwise, let **1** denote the vector of 1's in all coordinates, and define a new vector field $v^E$ on $\mathbb{R}^M$ as

$$v^E(x^E) := v_0^E(x^E) + \text{dist}^E(x^E, S^E)^2 \cdot \mathbf{1} \tag{B.24}$$

where $\text{dist}^E(x^E, S^E) := \inf_{y^E \in S^E} \| x^E - y^E \|$. Obviously, this new vector field agrees with $v_0^E(x^E)$ on $\mathcal{M}^E$ and therefore is still the pushforward of $v$ under $\iota$. Moreover, it is not uniformly 0 in a neighborhood of $S^E$. [It is worth noting that the so-defined vector field $v^E$ is in general *not* the (Euclidean) gradient of any function, a fact which presents significant difficulty to our analysis.]

**Step 2: $\mathcal{S}^E$ is an unstable invariant set.** Our next goal is to show that there exists an *unstable neighborhood* $\mathcal{U}^E$ around $\mathcal{S}^E$ in the following sense: First, for each $\hat{x} \in \mathcal{S}$, consider its image $\hat{x}^E = \iota(\hat{x}) \in \mathcal{S}^E$. Since $v^E$ agrees with the pushforward of $v \equiv \operatorname{grad} f$ under $\iota$, and since $\iota$ is an isometry, we have the following relation for all tangent vector $z \in \mathcal{T}_{\hat{x}}\mathcal{M}$:

$$\langle D v^E_{x^E} D \iota_{\hat{x}}(z), D \iota_{\hat{x}}(z) \rangle = \langle \operatorname{Hess} f(\hat{x})z, z \rangle_{\hat{x}}. \tag{B.25}$$

Since $\hat{x}$ is a strict saddle point, (B.25) implies that $\lambda_{\min}\left(D v^E_{\hat{x}^E}\right) < -c_- < 0$ for all $\hat{x}^E \in \mathcal{S}^E$. By an established series of arguments [10, 12, 46], using the stable manifold theorem for a strict saddle [63] and the transversality of the strict saddle manifold [1], there exists a $(M - m)$-dimensional embedded submanifold $\mathcal{Q}^E$ in $\mathbb{R}^M$ that contains $\mathcal{S}^E$. [Here, $1 \leq m \leq M$, and $M - m$ represents the dimension of the *unstable manifold* of $v^E$.] Moreover, writing $\Phi^E$ for the flow generated by $v^E$, it follows that $\mathcal{Q}^E$ is locally invariant under $\Phi^E$. Hence, there exists a neighborhood $\mathcal{N}^E$ of $\mathcal{S}^E$ in $\mathbb{R}^M$ and a positive time $t_0$ such that for all $|t| \leq t_0$, the following inclusion holds:

$$\Phi^E_t(\mathcal{N}^E \cap \mathcal{Q}^E) \subset \mathcal{Q}^E. \tag{B.26}$$

To proceed, note that $\mathbb{R}^M$ can be decomposed further as the direct sum of the tangent space to $\mathcal{Q}^E$ at $x^E$, denoted by $\mathcal{T}_{x^E}\mathcal{Q}^E$, and an additional complementary subspace denoted by $\mathcal{E}^u_{x^E}$:

$$\mathbb{R}^M = \mathcal{T}_{x^E}\mathcal{Q}^E \oplus \mathcal{E}^u_{x^E}. \tag{B.27}$$

The mapping $x^E \to \mathcal{E}^u_{x^E}$ is continuous, where $x^E$ varies over $\mathcal{S}^E$ and $\mathcal{E}^u_{x^E}$ belongs to the Grassmanian manifold $G(m, M)$. It is important to note that $\mathcal{E}^u_{x^E}$ *contains at least one direction in $\mathcal{T}_{x^E}\mathcal{M}^E$* due to (B.25). Then, for all $t \in \mathbb{R}$ and $x \in \mathcal{S}^E$, the Jacobian of $\Phi^E_t$ evaluated at $x^E$ maps $\mathcal{E}^u_{x^E}$ to $\mathcal{E}^u_{\Phi^E_t(x^E)}$, i.e.,

$$D \Phi^E_t(x^E) \mathcal{E}^u_{x^E} = \mathcal{E}^u_{\Phi^E_t(x^E)}. \tag{B.28}$$

Finally, and most importantly, we have the following characterization that formalizes the idea that *all directions in the unstable manifold should diverge under $\Phi^E_t$*: There exist positive constants $c$ and $C$ such that for all $x^E \in \mathcal{S}^E$, $w^E \in \mathcal{E}^u_{x^E}$, and $t \geq 0$, the following inequality holds:

$$\|D \Phi^E_t(x^E) w^E\| \geq C e^{ct} \|w^E\|. \tag{B.29}$$

The above verifies all the conditions for a *unstable invariant set* for $\mathcal{S}^E$ in the sense of Benaïm [10]. A deep result by Benaïm & Hirsch [11] then asserts the existence of a *local Lyapunov function* near a neighborhood of $\mathcal{S}^E$, whose construction we outline below.

**Step 3: Local Lyapunov function $\eta^E$.** For a right-differentiable function $\eta^E \colon \mathbb{R}^M \to \mathbb{R}$ we define its right derivative $D \eta^E$ applied to a vector $h^E \in \mathbb{R}^M$ by

$$D \eta^E(x^E) h^E = \lim_{t \to 0^+} \frac{\eta^E(x^E + t h^E) - \eta^E(x^E)}{t}. \tag{B.30}$$

If $\eta^E$ is differentiable, then (B.30) is simply $\langle \nabla \eta^E(x^E), h^E \rangle$. In view of all this, Benaïm [10] provides the following crucial result:

**Proposition B.1** (Benaïm, 1999, Prop. 9.5). *There exists a compact neighborhood $\mathcal{U}^E(\mathcal{S}^E)$ of $\mathcal{S}^E$, positive numbers $l, \beta > 0$, and a map $\eta^E \colon \mathcal{U}^E(\mathcal{S}^E) \to \mathbb{R}^+$ such that $\eta^E(x^E) = 0$ if and only if $x^E \in \mathcal{Q}^E$, and the following holds:*

  (i) *$\eta^E$ is $C^2$ on $\mathcal{U}^E(\mathcal{S}^E) \setminus \mathcal{Q}^E$.*
  (ii) *For all $x^E \in \mathcal{U}^E(\mathcal{S}^E) \cap \mathcal{Q}^E$, $\eta^E$ admits a right derivative $D \eta^E(x^E) \colon \mathbb{R}^M \to \mathbb{R}^M$ which is Lipschitz, convex and positively homogeneous.*
  (iii) *There exists $k > 0$ and a neighborhood $\mathcal{W}^E \subset \mathbb{R}^M$ of $0$ such that for all $x^E \in \mathcal{U}^E(\mathcal{S}^E)$ and $z^E \in \mathcal{W}^E$,*
  $$\eta^E(x^E + z^E) \geq \eta^E(x^E) + D \eta^E(x^E) z^E - k \|z^E\|^2. \tag{B.31}$$
  (iv) *There exists $c_1 > 0$ such that for all $x^E \in \mathcal{U}^E(\mathcal{S}^E) \setminus \mathcal{Q}^E$*
  $$\left\| \Pi_{\mathcal{M}^E} \left( D \eta^E(x^E) \right) \right\| \geq c_1 \tag{B.32}$$

*where $\Pi_{\mathcal{M}^E}$ is the projection on $\mathcal{M}^E$.[2] In addition, for all $x^E \in \mathcal{U}^E(\mathcal{S}^E) \cap \mathcal{Q}^E$ and $z^E \in \mathbb{R}^M$*

$$\langle \mathrm{D}\,\eta^E(x^E), z^E \rangle \geq c_1 \|z^E - \mathrm{D}\,\Pi(x^E)z^E\| \tag{B.33}$$

*where $\Pi$ is the projection of a neighborhood of $\mathcal{S}^E$ onto $\mathcal{Q}^E$.*

(v) *For all $x^E \in \mathcal{U}^E(\mathcal{S}^E) \cap \mathcal{Q}^E$, $w^E \in \mathcal{T}_{x^E}\mathcal{Q}^E$ and $z^E \in \mathbb{R}^M$,*

$$\mathrm{D}\,\eta^E(x^E)(w^E + z^E) = \mathrm{D}\,\eta^E(x^E)z^E. \tag{B.34}$$

(vi) *For all $x^E \in \mathcal{U}^E(\mathcal{S}^E)$ we have*

$$\mathrm{D}\,\eta^E(x)v^E(x^E) \geq \beta\eta^E(x^E). \tag{B.35}$$

The function $\eta^E$ will serve as a local "energy function" that plays an instrumental role in our analysis; the well-posedness of $\Pi$ is guaranteed by [27, Chap 4].

**Step 4: Geodesic offset.** Consider the image of an RRM scheme $X_n^E \coloneqq \iota(X_n)$. If $X_n^E \notin \mathcal{U}^E(\mathcal{S}^E)$ for all $n$, then there is nothing to prove. Otherwise, without loss of generality we may assume that $X_1^E \in \mathcal{U}^E(\mathcal{S}^E)$. Accordingly, define the first exit time $T$ from $\mathcal{U}^E(\mathcal{S}^E)$ as

$$T \coloneqq \inf\{k \geq 1 : X_n^E \notin \mathcal{U}^E(\mathcal{S}^E)\}. \tag{B.36}$$

Evidently, $T$ is a stopping time adaptive to $\mathcal{F}_n$, so it suffices to show that[3]

$$\mathbb{P}(T = \infty) = 0. \tag{B.37}$$

To this end, a notion that plays a central role in our analysis is the *geodesic offet*, defined as follows. Define the pushforward of the respective noise and bias vectors in the RRM scheme $X_n$ by

$$U_n^E \coloneqq \mathrm{D}\,\iota_{X_n}U_n, \quad b_n^E \coloneqq \mathrm{D}\,\iota_{X_n}b_n. \tag{B.38}$$

*It is important to remember that $U_n^E \in \mathcal{T}_{X_n^E}\mathcal{M}^E$, a fact that we will use freely in the sequel.*

We now formally define the *geodesic offset* $\Delta(x; z) \in \mathbb{R}^M$ as, for any $x \in \mathcal{M}$ and $z \in \mathcal{T}_x\mathcal{M}$,

$$\Delta(x; z) \coloneqq \iota(\exp_x(z)) - \iota(x) - \mathrm{D}\,\iota_x(z). \tag{B.39}$$

By Assumption 4, there exists $\varrho > 0$ such that, for all $\|z\|_x < \varrho$, the exponential mapping is the unique minimizing geodesic. Furthermore, for all such $z$'s, define the curve $\gamma^E(t) \coloneqq \iota(x) + t\,\mathrm{D}\,\iota_x(z)$, then

$$\gamma^E(0) = \iota(x), \quad \dot{\gamma}^E(0) = \mathrm{D}\,\iota_x(z) \tag{B.40}$$

so that $\gamma^E(t)$ agrees with the image of the geodesics $\iota(\exp_x(tz))$. As a result, for any $\|z\|_x < \varrho$, we have $\Delta(x; z) = \mathcal{O}(\|z\|_x^2)$. Now, setting $x \leftarrow X_n$ and $z \leftarrow \gamma_n(v(X_n) + U_n + b_n)$, we have

$$X_{n+1}^E = X_n^E + \gamma_n\big(v^E(X_n^E) + U_n^E + b_n^E\big) + \epsilon_n^E \tag{B.41}$$

where $\epsilon_n^E \coloneqq \Delta(X_n; \gamma_n(v(X_n) + U_n + b_n))$. By Assumption 3, we know that $U_n^E + b_n^E = \mathcal{O}(1)$ almost surely. Moreover, since $v$ is smooth, on the event $T = \infty$, $X_n^E \in \mathcal{U}^E(\mathcal{S}^E)$ for all $n$ and therefore $\sup_{n\geq 1}\|v^E(X_n^E)\| < \infty$. Since $\gamma_n \to 0$, for any $n$ large enough, we get

$$\epsilon_n^E = \mathcal{O}(\gamma_n^2). \tag{B.42}$$

Now, define two sequences of random variables $\{Y_n\}_{n\geq 1}$ and $\{S_n\}_{n\geq 1}$ as

$$Y_{n+1} = \big(\eta^E(X_{n+1}^E) - \eta^E(X_n^E)\big)\mathbb{1}_{\{n\leq T\}} + \gamma_n\mathbb{1}_{\{n>T\}}, \tag{B.43a}$$

$$S_0 = \eta^E(X_0^E), \quad S_n = S_0 + \sum_{k=1}^n Y_k. \tag{B.43b}$$

The importance of these sequences is that the event $\{T = \infty\}$ is contained in the event $\{S_n \to 0\}$. To see this, assume $T = \infty$. Then we have $Y_{n+1} = \eta^E(X_{n+1}^E) - \eta^E(X_n^E)$ and $S_n = \eta^E(X_n)$ by Eqs. (B.43a) and (B.43b). In addition, since $\{X_n^E\}$ remains in $\mathcal{U}^E(\mathcal{S}^E)$ by definition of the stopping time $T$, Theorem 2 combined with [10, Theorem 5.7] asserts that the limit set $L(\{X_n^E\})$ of $\{X_n^E\}$ is a nonempty compact invariant subset of $\mathcal{U}^E(\mathcal{S}^E)$, so that for all $y^E \in L(\{X_n^E\})$ and $t \in \mathbb{R}$, $\Phi_t^E(y^E) \in \mathcal{U}^E(\mathcal{S}^E)$. But then Proposition B.1(vi) implies that $\eta^E(\Phi_t^E(y^E)) \geq e^{\beta t}\eta^E(y^E)$ for all $t > 0$, forcing $\eta^E(y^E)$ to be zero. Since $\eta^E(x^E) = 0$ if and only if $x^E \in \mathcal{Q}^E$, we have $L(\{X_n^E\}) \subset \mathcal{Q}^E$, which implies $S_n = \eta^E(X_n^E) \to 0$.

Therefore, the rest of the proof is devoted to showing that $\mathbb{P}(\lim_{n\to\infty} S_n = 0) = 0$.

---

[2]The original statement in [10] is $\big\|\mathrm{D}\,\eta^E(x^E)\big\| \geq c_1$. The inequality in (B.32) is obtained via the same proof and noting that $\mathcal{E}_{x^E}^u$ contains at least one direction in $\mathcal{T}_{x^E}\mathcal{M}^E$.

[3]Theorem 1 follows from (B.37) because the event $\{\mathrm{dist}(X_n, \mathcal{S}) \to 0\}$ is contained in $\{T = \infty\}$.

**Step 5: Probabilistic estimates.** To this end, we will need two technical lemmas, originally due to Pemantle [55], and extended to their current form by Benaïm & Hirsch [12].

**Lemma B.1.** *Let $S_n$ be a nonnegative stochastic process, $S_n = S_0 + \sum_{k=1}^n Y_k$ where $Y_n$ is $\mathcal{F}_n$-measurable. Let $\alpha_n := \sum_{k=n}^\infty \gamma_k^2$. Assume there exist a sequence $0 \le \varepsilon_n = o(\sqrt{\alpha_n})$, constants $a_1, a_2 > 0$ and an integer $N_0$ such that for all $n \ge N_0$,*

- *(i) $|Y_n| = o(\sqrt{\alpha_n})$.*
- *(ii) $\mathbb{1}_{\{S_n > \varepsilon_n\}} \mathbb{E}[Y_{n+1}|\mathcal{F}_n] \ge 0$.*
- *(iii) $\mathbb{E}[S_{n+1}^2 - S_n^2 | \mathcal{F}_n] \ge a_1 \gamma_n^2$.*
- *(iv) $\mathbb{E}[Y_{n+1}^2 | \mathcal{F}_n] \le a_2 \gamma_n^2$.*

*Then $\mathbb{P}(\lim_{n\to\infty} S_n = 0) = 0$.*

**Lemma B.2.** *Let $S_n$ be a nonnegative stochastic process, $S_n = S_0 + \sum_{k=1}^n Y_k$ where $Y_n$ is $\mathcal{F}_n$-measurable and $|Y_n| \le C$ almost surely for some constant $C$. Assume that $\sum_n \gamma_n^2 = \infty$, and there exists $c > 0, N' \in \mathbb{N}$ such that for all $n \ge N'$,*

$$\mathbb{E}[S_{n+1}^2 - S_n^2 | \mathcal{F}_n] \ge c\gamma_n^2. \tag{B.44}$$

*Then*

$$\mathbb{P}\left(\lim_{n\to\infty} S_n = 0\right) = 0. \tag{B.45}$$

Our proof will be concluded by verifying all the premises of Lemmas B.1–B.2. To that end, note first that the sequence $S_n$ defined in (B.43b) is nonnegative by construction. We will then separate the analysis into two cases:

**Case 1.** *Square-summable step-sizes,* i.e., $\sum_n \gamma_n^2 < \infty$. In this case, we have

$$\lim_{n\to\infty} \frac{\gamma_n}{\sqrt{\sum_{k=n}^\infty \gamma_k^2}} = 0 \tag{B.46}$$

so $\gamma_n = o\left(\sqrt{\sum_{k=n}^\infty \gamma_k^2}\right)$. This fact will be used in the proof when we invoke Lemma B.1 below with $\varepsilon_n = \mathcal{O}(\gamma_n)$ and $\alpha_n = \sum_{k=n}^\infty \gamma_k^2$ therein. The verification process then proceeds as follows:

- **Verifying Lemma B.1(i) and (iv):** By the Lipschitz continuity of $\eta^E$, we know that

$$\begin{aligned}
\|\eta^E X_n^E - \eta^E X_{n+1}^E\| &\le L' \|X_n^E - X_{n+1}^E\| \\
&= \gamma_n \|v^E(X_n^E) + U_n^E + b_n^E\|
\end{aligned} \tag{B.47}$$

  where $L'$ is the Lipschitz constant of $\eta^E$. We have seen in the analysis of (B.42) that $\|v^E(X_n^E) + U_n^E + b_n^E\| = \mathcal{O}(1)$ almost surely by Assumption 3 on the event $T = \infty$. Therefore, $|Y_{n+1}| = \mathcal{O}(\gamma_n) = o(\sqrt{\alpha_n})$ which implies both Lemma B.1(i) and (iv).

- **Verifying Lemma B.1(ii):** Let $k' = k\|v^E\| + \sigma$ where $k$ is given by Proposition B.1(iii) and $\|v^E\| := \sup\{v^E(x^E) : x^E \in \mathcal{U}^E(\mathcal{S}^E)\}$ and $\sigma$ is the uniform bound of $U_n$. If $n \le T$, using Proposition B.1(ii), (iii), (v) and (vi) we have

$$\begin{aligned}
\eta^E(X_{n+1}^E) - \eta^E(X_n^E) &\ge \gamma_n \, \mathrm{D}\,\eta^E(X_n^E)\big(v^E(X_n^E) + U_n^E + b_n^E\big) \\
&\quad + \mathrm{D}\,\eta^E(X_n^E)\epsilon_n^E - k\gamma_n^2\big(\|v^E\| + \|U_n^E\| + \|b_n^E\|\big)^2 \\
&\ge \gamma_n \beta \eta^E(X_n^E) + \gamma_n \, \mathrm{D}\,\eta^E(X_n^E)U_n^E + \gamma_n \, \mathrm{D}\,\eta^E(X_n^E)b_n^E \\
&\quad + \mathrm{D}\,\eta^E(X_n^E)\epsilon_n^E - 2k'\gamma_n^2 - 2k\gamma_n^2\|b_n^E\|^2.
\end{aligned} \tag{B.48}$$

  By Assumption 3, there exists a constant $c' > 0$ such that $-\|b_n^E\| \ge -c'\gamma_n$ (a.s.). Combining this with the Lipschitz continuity of $\eta^E$ and (B.42), we can merge the last four terms in (B.48) as

$$\eta^E(X_{n+1}^E) - \eta^E(X_n^E) \ge \gamma_n \beta \eta^E(X_n^E) + \gamma_n \, \mathrm{D}\,\eta^E(X_n^E)U_n^E - 2k''\gamma_n^2 \tag{B.49}$$

  for some constant $k'' > 0$. We thus get

$$\mathbb{1}_{\{n\le T\}} \mathbb{E}[Y_{n+1}|\mathcal{F}_n] \ge \mathbb{1}_{\{n\le T\}}\big[\gamma_n \beta \eta^E(X_n^E) - 2k''\gamma_n^2 + \gamma_n \mathbb{E}[\mathrm{D}\,\eta^E(X_n^E)U_n^E|\mathcal{F}_n]\big]. \tag{B.50}$$

By Proposition B.1(ii) again, we have

$$\mathbb{E}[\mathrm{D}\,\eta^E(X_n^E)U_n^E|\mathcal{F}_n] \geq \mathrm{D}\,\eta^E(X_n^E)\,\mathbb{E}[U_n^E|\mathcal{F}_n] = 0$$
$$= \mathrm{D}\,\eta^E(X_n^E)\,\mathbb{E}[\mathrm{D}\,\iota_{x_n}U_n|\mathcal{F}_n] = 0 \tag{B.51}$$

since we have assumed the noise to be zero mean. Combining (B.50) and (B.51), we then get

$$\mathbb{1}_{\{n\leq T\}}\,\mathbb{E}[Y_{n+1}|\mathcal{F}_n] \geq \mathbb{1}_{\{n\leq T\}}\big[\gamma_n\beta\eta^E(X_n^E) - 2k''\gamma_n^2\big]. \tag{B.52}$$

If $n > T$, $Y_{n+1} = \gamma_n$ so trivially

$$\mathbb{1}_{\{n\leq T\}}\,\mathbb{E}[Y_{n+1}|\mathcal{F}_n] \geq 0. \tag{B.53}$$

Combining (B.52) with (B.53), we see that Lemma B.1(ii) is satisfied with $\varepsilon_n = \frac{k''}{\beta}\gamma_n$.

- **Verifying Lemma B.1(iii):** We begin by observing that

$$\mathbb{E}[S_{n+1}^2 - S_n^2|\mathcal{F}_n] = \mathbb{E}[Y_{n+1}^2|\mathcal{F}_n] + 2S_n\,\mathbb{E}[Y_{n+1}|\mathcal{F}_n]. \tag{B.54}$$

If $S_n \geq \varepsilon_n$, then the last term on the right-hand side of (B.54) is non-negative by Lemma B.1(ii) that we just verified above. If $S_n < \varepsilon_n$, (B.52) with (B.53) imply that $S_n\,\mathbb{E}[Y_{n+1}|\mathcal{F}_n] \geq -\varepsilon_n k''\gamma_n^2 = -\mathcal{O}(\gamma_n^3)$. In other words, (B.54) can be rewritten as

$$\mathbb{E}[S_{n+1}^2 - S_n^2|\mathcal{F}_n] \geq \mathbb{E}[Y_{n+1}^2|\mathcal{F}_n] - \mathcal{O}(\gamma_n^3). \tag{B.55}$$

Below, we shall prove that $\mathbb{E}[Y_{n+1}^2|\mathcal{F}_n] \geq b_1\gamma_n^2$ for some $b_1 > 0$ and $n$ large enough. Combining this with (B.55) proves Lemma B.1(iii).

From (B.49), we deduce

$$\mathbb{1}_{\{n\leq T\}}\Big[\mathbb{E}[(Y_{n+1})_+|\mathcal{F}_n] - \big(\gamma_n\,\mathbb{E}[(\mathrm{D}\,\eta^E(X_n^E)U_n^E)_+|\mathcal{F}_n] - k''\gamma_n^2\big)\Big] \geq 0. \tag{B.56}$$

We now claim that

$$\mathbb{1}_{\{n\leq T\}\cap\{X_n^E\notin\mathcal{Q}^E\}}\Big(\mathbb{E}[(\mathrm{D}\,\eta^E(X_n^E)U_n^E)_+|\mathcal{F}_n]\Big) \geq c_1\zeta. \tag{B.57}$$

where $c_1$ is given by Proposition B.1(iv) and $\zeta$ is defined in Assumption 3. To see this, recall that $\|U_n\|_{x_n} < \sigma$ by Assumption 3. Moreover, we have $X_n^E \in \mathcal{U}^E(\mathcal{S}^E)$ on the event $T = \infty$. Proposition B.1(i) then implies $\eta^E$ is differentiable on $\mathcal{U}^E(\mathcal{S}^E) \setminus \mathcal{Q}^E$, and Proposition B.1(iv) further shows that

$$\mathbb{1}_{\{n\leq T\}\cap\{X_n^E\notin\mathcal{Q}^E\}}\Big(\mathbb{E}[(\mathrm{D}\,\eta^E(X_n^E)U_n^E)_+|\mathcal{F}_n]\Big)$$
$$= \mathbb{1}_{\{n\leq T\}\cap\{X_n^E\notin\mathcal{Q}^E\}}\Big(\mathbb{E}[[\langle\eta^E(X_n^E), U_n^E\rangle]_+|\mathcal{F}_n]\Big)$$
$$= \mathbb{1}_{\{n\leq T\}\cap\{X_n^E\notin\mathcal{Q}^E\}}\Big(\mathbb{E}\big[[\langle\Pi_{\mathcal{T}_{X_n^E}\mathcal{M}^E}(\eta^E(X_n^E)), U_n^E\rangle]_+|\mathcal{F}_n\big]\Big)$$
$$\geq c_1\zeta \tag{B.58}$$

where we have used the fact that $U_n^E \in \mathcal{T}_{X_n}\mathcal{M}^E$ and Assumption 3.

If $X_n^E \in \mathcal{Q}^E$, we can choose a unit vector $v_n^E \in \ker(I - \mathrm{D}\,\Pi(X_n^E))^\perp \cap \mathcal{T}_{X_n^E}\mathcal{M}^E$ where $\Pi$ denotes the projection operator onto $\mathcal{Q}^E$; note that $v_n^E \in \ker(I - \mathrm{D}\,\Pi(X_n^E))^\perp \cap \mathcal{T}_{X_n^E}\mathcal{M}^E \neq \emptyset$ since $\mathcal{E}_{x^E}^u$ contains at least one direction in $\mathcal{T}_{x^E}\mathcal{M}^E$ for all $x^E \in \mathcal{U}^E(\mathcal{S}^E) \cap \mathcal{M}^E$. By the definition of $v_n^E$, we have $\langle U_n^E, v_n^E\rangle = \langle U_n^E - \mathrm{D}\,\Pi(X_n^E)U_n^E, v_n^E\rangle$. Let $\mathcal{H} = \{n \leq T\} \cap \{X_n^E \in \mathcal{Q}^E\}$. By Proposition B.1(iv), Cauchy-Schwartz, and Assumption 3, we get

$$\mathbb{1}_{\mathcal{H}}\,\mathbb{E}\big[[\mathrm{D}\,\eta^E(X_n^E)U_n^E]_+|\mathcal{F}_n\big] \geq c_1\,\mathbb{1}_{\mathcal{H}}\,\mathbb{E}\big[\|U_n^E - \mathrm{D}\,\Pi(X_n^E)U_n^E\||\mathcal{F}_n\big]$$
$$\geq c_1\,\mathbb{1}_{\mathcal{H}}\,\mathbb{E}\big[[\langle U_n^E - \mathrm{D}\,\Pi(X_n^E)U_n^E, v_n^E\rangle]_+|\mathcal{F}_n\big]$$
$$= c_1\,\mathbb{1}_{\mathcal{H}}\,\mathbb{E}\big[[\langle U_n^E, v_n^E\rangle]_+|\mathcal{F}_n\big]$$
$$= c_1\,\mathbb{1}_{\mathcal{H}}\,\mathbb{E}\big[[\langle U_n, v_n\rangle_{x_n}]_+|\mathcal{F}_n\big] \tag{B.59}$$

where $v_n$ is the *pullback* of $v_n^E$ under $\iota$. Since $\iota$ is an isometry, the pullback preserves the inner product, and therefore

$$\mathbb{1}_{\mathcal{H}}\,\mathbb{E}\big[[\mathrm{D}\,\eta^E(X_n^E)U_n^E]_+|\mathcal{F}_n\big] \geq c_1\zeta\,\mathbb{1}_{\mathcal{H}} \tag{B.60}$$

by Assumption 3. Combining Eqs. (B.53), (B.56) and (B.60) and Item **Case 1.** then gives

$$\mathbb{E}[[Y_{n+1}]_+|\mathcal{F}_n] \geq c_1\zeta\gamma_n - k''\gamma_n^2. \tag{B.61}$$

On the other hand, we always have $\mathbb{E}[Y_{n+1}^2|\mathcal{F}_n] \geq \mathbb{E}[[Y_{n+1}]_+|\mathcal{F}_n]^2$ by Jensen. It then follows that $\mathbb{E}[Y_{n+1}^2|\mathcal{F}_n] \geq b_1\gamma_n^2$ for some $b_1 > 0$ and large enough $n$ as desired.

We have now verified conditions (i)–(iv) in Lemma B.1. Thus, Lemma B.1 concludes that

$$\mathbb{P}(\lim_{n\to\infty} S_n = 0) = 0 \tag{B.62}$$

which finishes the proof for the case of $\sum_n \gamma_n^2 < \infty$.

**Case 2.** When $\sum_n \gamma_n^2 = \infty$, the same proof above shows that $\mathbb{E}[Y_{n+1}^2|\mathcal{F}_n] \geq b_1\gamma_n^2$ for some $b_1 > 0$ and large enough $n$. Combining this with (B.55) yields

$$\mathbb{E}[S_{n+1}^2 - S_n^2|\mathcal{F}_n] \geq c\gamma_n^2 \tag{B.63}$$

for some $c > 0$. Lemma B.2 then concludes:

$$\mathbb{P}(\lim_{n\to\infty} S_n = 0) = 0$$

as claimed. ∎

## B.3. Proof of Proposition 1.

We conclude this appendix with the application of Theorem 1 to Algorithms 1–5 under the explicit oracle assumptions of Proposition 1. For convenience, we restate the relevant result below:

**Proposition 1.** *Suppose that Algorithms 1–5 are run with a gradient oracle $V(x;\theta) = v(x) + \mathrm{err}(x;\theta)$ such that*

$$\|\mathrm{err}(x;\theta)\|_x \leq \sigma(x) \quad and \quad \mathbb{E}[[\langle\mathrm{err}(x;\theta), z\rangle_x]_+] \geq \zeta(x) \tag{17}$$

*for all $z \in \mathcal{T}_x\mathcal{M}$, $\|z\|_x = 1$, and for suitable functions $\zeta, \sigma \colon \mathcal{M} \to \mathbb{R}_+$ with $\sigma$ bounded on bounded subsets of $\mathcal{M}$ and $\inf_x \zeta(x) > 0$. Then, under Assumptions 2 and 4, the conclusion of Theorem 1 holds, that is, Algorithms 1–5 avoid strict saddle manifolds of $f$.*

*Remark.* In additional to the claimed Algorithms 1–5, we will further prove the same conclusion for the two algorithms considered in Appendix A.

*Proof.* By Theorem 1, it suffices to verify Assumption 3 under (17) and the event $\mathrm{dist}(X_n, \mathcal{S}) \to 0$. We proceed method by method.

**Algorithm 1.** Since $b_n = 0$ in Algorithm 1, Assumption 3 holds trivially by (17).

**Algorithms 2, 3 and 5.** By definition, $\mathcal{R}_x(z)$ is a smooth map and hence satisfies $\lim_{z\to 0} \mathcal{R}_x(z) = x$. On the event $\mathrm{dist}(X_n, \mathcal{S}) \to 0$, we have $v(X_n) + U_n + b_n = \mathcal{O}(1)$, and therefore $X_{n+1}$ lies in the injectivity radius of $X_n$ with probability 1 for $n$ large enough. As a result, the mapping $\log_{X_n}(X_{n+1})$ is well-define for all $n$ large enough.

We first consider Algorithms 2 and 5 whose proofs are identical since they are both are the form:

$$X_{n+1} = \mathcal{R}_{X_n}(\gamma_n V(X_n;\theta_n)). \tag{B.64}$$

Let $\tilde{v}_n \in \mathcal{T}_{X_n}\mathcal{M}$ be the vector such that $\exp_{X_n}(\gamma_n\tilde{v}_n) = X_{n+1}$, i.e.,

$$\gamma_n\tilde{v}_n = \log_{X_n}\left(\mathcal{R}_{X_n}(\gamma_n V(X_n;\theta_n))\right). \tag{B.65}$$

Then (B.64) is an RRM scheme with $W_n = \tilde{v}_n - v(X_n)$ where $\tilde{v}_n$ is defined in (B.65). Consider the curve $c(t) := \mathcal{R}_{X_n}(tV(X_n;\theta_n))$. By (17), on the event $\mathrm{dist}(X_n, \mathcal{S}) \to 0$, the curve $c(t)$ lies in the injectivity radius of $X_n$ almost surely for all $t \in [0, \gamma_n]$ and all $n$ large enough. Let $\hat{c}(t)$ be the smooth curve of $c(t)$ in the normal coordinate with base $X_n$ and an arbitrary orthonormal frame, and let $\hat{X}_{n+1}$ be the normal coordinate of $X_{n+1}$. Also, let $\tilde{v}_n^N$ be the (Euclidean) vector of $\tilde{v}_n$ expanded in the chosen orthonormal basis, and define $V^N(X_n;\theta_n)$ and $\mathrm{err}^N(X_n;\theta_n)$ similarly. By definition, $\hat{X}_{n+1}$ is nothing but $\gamma_n\tilde{v}_n^N$. In addition, by (17), we have

$$\|\mathrm{err}^N(X_n;\theta_n)\| = \|\mathrm{err}(X_n;\theta_n)\|_{X_n} \leq \sigma \tag{B.66}$$

for some $\sigma < \infty$.

Since $X_n = c(0)$ and $X_{n+1} = c(\gamma_n)$, by properties of a retraction map we must have

$$
\begin{aligned}
\gamma_n \tilde{v}_n^{\mathrm{N}} &= \hat{c}(\gamma_n) \\
&= \hat{c}(0) + \gamma_n \dot{\hat{c}}(0) + \mathcal{O}\Big(\gamma_n^2 \|\dot{\hat{c}}(0)\|_2^2\Big) \\
&= \gamma_n V^{\mathrm{N}}(X_n; \theta_n) + \mathcal{O}\Big(\gamma_n^2 \|V(X_n; \theta_n)\|_{X_n}^2\Big) \\
&=: \gamma_n V^{\mathrm{N}}(X_n; \theta_n) + \gamma_n \tilde{b}_n
\end{aligned}
\tag{B.67}
$$

where $\tilde{b}_n = \mathcal{O}\big(\gamma_n \|V(X_n; \theta_n)\|_{X_n}^2\big) = \mathcal{O}(\gamma_n)$. Therefore,

$$
\|b_n\|_{X_n} = \|\mathbb{E}[W_n \mid \mathcal{F}_n]\|_{X_n} = \big\|\mathbb{E}\big[\tilde{b}_n \mid \mathcal{F}_n\big]\big\| = \mathcal{O}(\gamma_n)
\tag{B.68}
$$

which proves the condition for $b_n$ in Assumption 3. On the other hand, (B.67) shows that

$$
\begin{aligned}
\|U_n\|_{X_n} &\leq \|V^{\mathrm{N}}(X_n; \theta_n) + \tilde{b}_n\| + \|\mathbb{E}\big[V^{\mathrm{N}}(X_n; \theta_n) + \tilde{b}_n\big]\| \\
&= \mathcal{O}(1)
\end{aligned}
$$

since $\|V^{\mathrm{N}}(X_n; \theta_n)\| = \mathcal{O}(1)$ by (17) and $\tilde{b}_n = \mathcal{O}(\gamma_n)$. Finally, for any unit vector $z \in \mathcal{T}_{X_n}\mathcal{M}$, (B.67) implies

$$
\begin{aligned}
\mathbb{E}[[\langle z, U_n \rangle_{X_n}]_+] &\geq \mathbb{E}[[\langle z, \mathrm{err}(X_n; \theta_n) \rangle_{X_n}]_+] - \|\tilde{b}_n\| \\
&= \mathbb{E}[[\langle z, \mathrm{err}(X_n; \theta_n) \rangle_{X_n}]_+] - \mathcal{O}(\gamma_n).
\end{aligned}
\tag{B.69}
$$

Since $\gamma_n \to 0$, this finishes the proof of Algorithms 2 and 5. For Algorithm 3, an Euclidean oracle of the form (17) translates to a Riemannian oracle with $\mathrm{err}'(x; \theta) := \nabla^2 h(x)^{-1} \mathrm{err}(x; \theta)$. It then suffices to note that, on the event $\mathrm{dist}(X_n, \mathcal{S}) \to 0$, $\nabla^2 h(X_n)$ is both upper and lower bounded.

**Algorithms 4, 6 and 7.** For (RSEG), $U_n = \Gamma_{X_n^+ \to X_n}(\mathrm{err}(X_n^+; \theta_n^+))$ so

$$
\|U_n\|_{X_n} = \big\|\mathrm{err}(X_n^+; \theta_n^+)\big\|_{X_n^+} \leq \sigma
\tag{B.70}
$$

by (17) and the fact that the parallel transport map is a linear isometry. For the bias term, the definition of (RSEG) yields

$$
\|b_n\| = \|\Gamma_{X_n^+ \to X_n}(v(X_n^+)) - v(X_n)\|_{X_n} \leq L \, \mathrm{dist}(X_n^+, X_n) = \gamma_n L \|V(X_n; \theta_n)\|_{X_n} = \mathcal{O}(1)
\tag{B.71}
$$

by the same argument as for Algorithms 2, 3 and 5.

For (ROG), we have $U_n = \Gamma_{X_n^+ \to X_n}(\mathrm{err}(X_n; \theta_n^+))$ and $b_n = \Gamma_{X_n^+ \to X_n}(v(X_n^+)) - v(X_n)$, so Assumption 3 can be checked exactly as in the case of Algorithm 7 above. The analysis for Algorithm 6 is similar so we omit the details. ∎

