# OpenReview forum: "Riemannian stochastic optimization methods avoid strict saddle points"
_NeurIPS.cc/2023/Conference — NeurIPS 2023 poster_

### Official Review · Reviewer_K9Za · 2023-07-02

**Soundness:** 4 excellent
**Presentation:** 4 excellent
**Contribution:** 3 good
**Rating:** 8
**Confidence:** 3

**Summary:**

The authors study saddle point avoidance of stochastic gradient algorithms formulated on Riemannian manifolds. They prove a number of results on saddle avoidance for which the Euclidean analogs, possibly under slightly different technical assumptions, are known. These results show that stochastic gradient descent algorithms also avoid strict saddle points even if working in a geometric setting. This is a purely theoretical paper and does not include any experiments or empirical examples.

EDIT: I've read the rebuttal, and am happy to keep my score where it was.

**Strengths:**

This paper addresses an important topic and is very-well-written, particularly its main text. It is very readable to those with theoretical training, but not necessarily experts in optimization - a common pitfall among works written at a similar technical level.

Due to field's historical emphasis on convex optimization at the expense of non-convex optimization, theory on saddle point avoidance is only beginning to become available. At the same time, Riemannian optimization is a rapidly growing area, and has the potential to help bring improved understanding to optimization at a time when doing so is increasingly important due to the rise of deep learning. Therefore, the contribution is timely and valuable

The technical appendix reads as if it was written carefully. I spotted no hints of sloppiness that would call the results into question, but at the same time this work is heavy enough that typos - including potentially in references the work relies on - are a concern. I did not do a full check, since my primary work is in a different (albeit related) area, and this paper is technical enough that it needs an expert familiar other saddle-avoidance papers for the verification to be trustworthy. I encourage the authors to do a careful read-over after not thinking about this paper for at least a month, so as to go over the logic and spot typos. With this caveat stated, the paper clearly meets the NeurIPS bar from a technical viewpoint.

The introduction is particularly well put together, makes clear what is already done, what the authors are adding, and what the key takeaways are. I would use this paper as an example to students for how to write a good introduction.

**Weaknesses:**

No experiments. This paper would have been made significantly stronger with even a minimal empirical study to illustrate how a few examples of RSGD behave on common manifolds. It would have been particularly interesting to see a case or two where RSGD does not get stuck, confirming the theory, and one which violates the theory's assumptions where it does get stuck, illustrating the limitations.

Sec 2: consider a more informative title

Optimization-theoretic assumptions involving the setting should be stated more clearly clearly all in one location, so that optimization theory people can easily see how strong the obtained results are:
* Lipschitz gradient
* Avoidance defined in terms of strict saddle points
* Geodesically complete manifold
* Maybe consider adding \paragraph called "Key Assumptions"?

Paper takes a bit too long to get to the point, page 5 is still describing the setting. Even though the description is good, consider simplifying by having less examples:
* Consider splitting the examples into one or two key examples, then have the results more in the middle of the paper, then having a final section which describes additional examples

The paper talks way too many times about how intricate the analysis is. My experience is that most publishable forms of analysis are either intricate or they are obvious, so stating this just makes it feel like the authors want to make themselves look fancy and important to the readers. This is distracting and takes away from the paper's actual content. Rather than saying it's intricate, maybe instead say what specific elements of the analysis make it so.

**Questions:**

Certain points that describe what is new could be given in more detail:
* Line 336: why does this work's probabilistic analysis diverge significantly?

Step 2: would it be fair to say this part of the proof is effectively via a reduction to the Euclidean case?

General question: why is the defined "(strict) saddle manifold" actually a manifold? If the saddle points are mere points, and not ridges, this'll (trivially) be a zero-dimensional manifold, as it is a finite or countable point set, but what if there are ridges? Does something assumption here that I've missed prevent singularities from appearing? May help to include some intuition here.

Am I right to think of the RRM scheme as traveling some distance in a tangent space, then mapping back onto the manifold? Thought in this way, it starts to feel a bit like mirror descent, which also projects back onto the space.

Am I right to think about the retraction version as "viewing the difference between the retraction and exponential map as noise, which we then simply merge with the gradient noise"?

Typos:
* 48: "scopein" -> typo
* Line 626: wrong \eqref, this should instead refer to (13), not to (12)
* Appendix B: title in the PDF table of contents is "Proof of thm:repeller", use \texorpdfstring to fix this (you can look up what it does)

**Limitations:**

Yes, this is a theory paper and the technical assumptions are clearly stated throughout

---

> ### Author Rebuttal · Authors · 2023-08-08
>
> We are sincerely grateful for your encouraging remarks and thoughtful suggestions. We reply to your questions below, and we will revise our paper accordingly in the next revision opportunity.
>
>
> > This paper would have been made significantly stronger with even a minimal empirical study to illustrate how a few examples of RSGD behave on common manifolds. It would ha been particularly interesting to see a case or two where RSGD does not get stuck, confirming the theory, and one which violates the theory's assumptions where it does get stuck, illustrating the limitations.
>
> We took your point to heart and we generated a series of plots to illustrate how RSGD avoids saddle points on the torus (everything is included in the one-page pdf that we were allowed to upload). Regarding your second point, the stochasticity involved makes it quite difficult to construct an example where RSGD does indeed get stuck with positive probability but, on the flip side, we performed a series of experiments with the extra-gradient method RSEG to illustrate that our avoidance results are not limited to RSGD.
>
>
> > Sec 2: consider a more informative title.
>
> Will do - we will highlight that it is intended to fix some basic definitions of Riemannian geometry and optimization.
>
>
> > Optimization-theoretic assumptions involving the setting should be stated more clearly all in one location, so that optimization theory people can easily see how strong the obtained result
>
> Will do - we will modify the beginning of Section 4 accordingly, along the lines you recommended.
>
>
> > Consider splitting the examples into one or two key examples, then have the results more in the middle of the paper, then having a final section which describes additional examples.
>
> Point taken. We did want to emphasize the flexibility afforded by the RRM template, but we also understand the benefit of getting to the point faster. We will restructure accordingly (sending some examples to the appendix if needed).
>
>
> > The paper talks way too many times about how intricate the analysis is.
>
> Apologies, we did not mean to make the exposition sound bloated or over-inflated. We will remove all such instances that we spot, and we will do a full editorial pass to remove any adverbs and/or adjectives that are subjective.
>
>
> > Line 336: why does this work's probabilistic analysis diverge significantly?
>
> This is due to the disparities in the step-size and noise assumptions: To provide a simplified overview, an essential step in the analysis is to control the fluctuation of cumulative noise $\sum_{n\geq m} \gamma_nU_n$, where $\gamma_n$ is the step-size and $U_n$ is the noise. In [31] (cf. numbering as in the full paper, including the appendix), because the step-size schedule exhibits a significantly faster decrease rate $\Omega(1/\sqrt{n})$, the Burkholder-Davis-Gundy inequality serves as the appropriate tool for that scenario. However, this analysis is inadequate when dealing with gradually decreasing step-sizes of the form $\mathcal{O}(1/(\log n)^{1+\epsilon})$ in our particular context.
>
>
> >Step 2: would it be fair to say this part of the proof is effectively via a reduction to the Euclidean case?
>
> Yes, this is the driving idea.
>
>
> >General question: why is the defined "(strict) saddle manifold" actually a manifold? If the saddle points are mere points, and not ridges, this'll (trivially) be a zero-dimensional manifold, as it is a finite or countable point set, but what if there are ridges? Does something assumption here that I've missed prevent singularities from appearing? May help to include some intuition here.
>
> Apologies for any confusion. When we wrote "smooth compact component" in the definition of a strict saddle manifold, the "smooth" was meant in the manifold sense. As you recommended above, we will bring this definition in Section 4 to collect all relevant information in a single place and ensure there is no ambiguity or confusion.
>
>
> >Am I right to think of the RRM scheme as traveling some distance in a tangent space, then mapping back onto the manifold? Thought in this way, it starts to feel a bit like mirror descent, which also projects back onto the space.
>
> This is not our intuition. For us, RRM is traveling on the manifold directly, along a search direction defined by an "approximate" / "surrogate" gradient, including noise and/or a (possibly non-random) bias/offset. This bias/offset term is crucial for our purposes as it allows us to capture several different algorithms, possibly with a forward-backward-forward structure (like ROG and RSEG). However, the RRM iterates per se never leave the underlying manifold.
>
> When embedded in real space, the part $X_n + \gamma_n \hat V_n$ of (16) which represents the RRM update minus the geodesic offset does exhibit the behavior that you describe. If this is what you are referring to, then we agree with your intuition - with the "geodesic offset" essentially standing in for the projection mechanism.
>
>
> >Am I right to think about the retraction version as "viewing the difference between the retraction and exponential map as noise, which we then simply merge with the gradient noise"?
>
> Your intuition is accurate, with a minor caveat. The gradient "noise" is conventionally assumed to have zero mean, while the difference between the retraction and exponential map does not possess this property. To circumvent any potential confusion, we opt for the term "error" instead of "noise" when discussing the differences between random and systematic errors.
>
> > {Typos}
>
> Will fix, thanks for spotting them and bringing them to our attention!
>
> ---
>
> We hope that these points address your questions - please let us know if any of the above is not sufficiently clear.
>
> Thank you again for your input and positive evaluation,
>
> The authors

---

> > ### Comment · Reviewer_K9Za · 2023-08-14
> >
> > Thanks for your response. I continue to like this paper and am happy for it to be accepted.

---

> > > ### Author Response · Authors · 2023-08-20
> > > **Official Comment by Authors**
> > >
> > > Thank you for the in-depth review and encouraging comments.

---

### Official Review · Reviewer_SLQm · 2023-07-03

**Soundness:** 4 excellent
**Presentation:** 4 excellent
**Contribution:** 3 good
**Rating:** 7
**Confidence:** 3

**Summary:**

This paper proves that under a rather general RRM scheme (akin to vanilla GD in euclidean space), strict saddles can be avoided when stochastic approaches are used.

**Strengths:**

1. Proves that Riemannian Optimization, much like its euclidean counterpart, can have its strict saddles easily avoided by using a simple update rule (RRM in this case) coupled with perturbations.

2. This paper analyzed a number of different stochastic algorithms (algorithms 1-5) and shows that they all can escape saddle points.

3. The proof outline is nice. It provides readers a new proof technique that can deal with Riemanian optimization.



**Weaknesses:**

1. For unfamiliar readers, it is hard to gauge how limiting the "our assumptions 1-3" are, and I would like to see the authors provide a bit more analysis on the limitations of these assumptions.

2. Maybe it's better to include the convergence analysis of the RRM update rule to let users understand better the limitations of this result (it doesn't need to be self-derived, it's perfectly fine to quote well-known results).

**Questions:**

I want to know how important the noise perturbation is to this analysis. For the RRM, if no random error is observed, will the theorems still work? Also I want to know whether random perturbations is more important for saddle-escaping or the stochastic oracles are more important.

---

> ### Author Rebuttal · Authors · 2023-08-08
>
> Thank you for your input, insightful questions, and positive evaluation. We address each of your questions in a point-by-point thread below, and we will revise our manuscript accordingly in the next revision opportunity.
>
> >For unfamiliar readers, it is hard to gauge how limiting the "our assumptions 1-3" are, and I would like to see the authors provide a bit more analysis on the limitations of these assumptions.
>
> We understand your concern. It is for this reason that we had included a specific paragraph for discussing each assumption in detail but, at the same time, we were constrained by the 9-page limit and could not include a more extensive positioning for unfamiliar readers. We will be happy to take advantage of the extra page afforded in the upcoming revision to expand on our discussion and provide more details for readers less familiar with the relevant literature.
>
>
> >Maybe it's better to include the convergence analysis of the RRM update rule to let users understand better the limitations of this result (it doesn't need to be self-derived, it's perfectly fine to quote well-known results).
>
> Point well-taken. Again, due to space limitations, we had to make some tough choices in terms of what to present and what to leave out, but we will be happy to take advantage of the extra page provided in the revision stage to include the basic steps of the convergence analysis for RRM.
>
>
> >I want to know how important the noise perturbation is to this analysis. For the RRM, if no random error is observed, will the theorems still work?
>
> We did not address the noiseless scenario because, to a certain extent, it had already been examined in various prior studies for RGD [34,35]. The high-level conclusion is that, depending on the initialization, Riemannian first-order methods may or may not evade saddle points; however, the set of initial conditions that result in convergence to saddle points occupies a Lebesgue measure of 0, aligning conceptually with our result that $\mathbb{P}(X_n \rightarrow \mathcal{S}) = 0$ for stochastic methods.
>
> It is noteworthy to mention that the analysis of the noiseless scenario follows a straightforward trajectory: it emerges as a direct outcome of linearization near saddle points and the implicit function theorem. However, this approach is not sufficient in the presence of noise, leading to a substantially more intricate proof.
>
>
> >Also I want to know whether random perturbations is more important for saddle-escaping or the stochastic oracles are more important.
>
> The direction the Reviewer is seeking to emphasize would determine the context: Our paper asserts that, in the presence of stochastic oracle feedback, *no random perturbations* are needed to avoid saddle points. Previous research indicates that if the primary objective is to elude saddle points *efficiently*, then random perturbations can be employed to accelerate the process of escape - but, otherwise, in this "efficient escape" literature, the optimizer is assumed to have full access to the function's gradients.
>
> ---
>
> We hope and trust that these points address your questions - but please let us know if any of the above is not sufficiently clear.
>
> Thank you again for your input and positive evaluation,
>
> The authors

---

> > ### Comment · Reviewer_SLQm · 2023-08-16
> >
> > Thank you for your response, as it cleared some of my questions, not necessarily concerns of your paper. I continue to believe in the value in this work, and would like to keep my original score.

---

> > > ### Author Response · Authors · 2023-08-20
> > > **Official Comment by Authors**
> > >
> > > Thank you for the encouraging feedback and helpful comments.

---

### Official Review · Reviewer_5i8d · 2023-07-06

**Soundness:** 3 good
**Presentation:** 3 good
**Contribution:** 3 good
**Rating:** 7
**Confidence:** 4

**Summary:**

The paper presents a focused study on the avoidance of saddle points for stochastic Riemannian optimization algorithms. To tackle this issue, the authors introduce the Riemannian Robbins-Monro (RRM) schemes to the context of Riemannian manifolds, where includes fundamental Riemannian stochastic optimization methods as special cases, such as Riemannian stochastic Gradient Descent (GD), Riemannian proximal GD, and Riemannian optimistic GD.

In their analysis, the authors have modeled the RRM within the context of a Euclidean stochastic approximation framework by famous Nash embedding theorem, introducing an element of geodesic offset. The authors make the assumption that this offset is $\mathcal O(\gamma_n^2)$, a condition found to be met in the case of a bounded-variance stochastic gradient estimator as indicated in Proposition 1. Based on this assumption, they have been able to indicate that RRM schemes have the capacity to avoid saddle points with a probability of 1.


**Strengths:**

- Originality: The analysis presented in Theorem 1 and Theorem 2 is highly innovative. In the analysis, the author apply the Nash embedding theorem to isometrically embed the Riemannian manifold into a Euclidean space. This approach allows them to approximate the intrinsic geometry of the manifold using its extrinsic (now Euclidean) geometry, an approach that seems highly original and may potentially shed light on further development in Riemannian optimization methods.

- Quality: The quality of the paper is commendable. The paper is well-written, and the results are convincing.

- Clarity: The paper is well-structured. The assumptions made are either standard in the field of Euclidean optimization or are accompanied by discussions regarding their implications,  making the paper easy to follow for readers.

- Significance: The paper is the first to generalize saddle point avoidance for stochastic optimization methods on Riemannian manifolds. Additionally, the proposed Riemannian Robbins-Monro (RRM) schemes encompass numerous state-of-the-art Riemannian stochastic optimization algorithms, illustrating potential applications in tackling Riemannian machine learning problems.

**Weaknesses:**

- Possible typo: l52 shcemes -> schemes.

- It would be better if there are numerical experiments to validate the findings.

**Questions:**

- In the conclusion section of your paper, you mention 'zeroth-order optimization' as a potential avenue for future work. There have been several studies related to the zeroth-order (or so-called 'bandit') gradient descent method ([1,2]). I am interested to know whether such zeroth-order methods fit within your proposed Riemannian Robbins-Monro (RRM) scheme. If not, could you please clarify the primary gap or obstacle in incorporating these methods into the RRM framework?

- A lot of work on Riemannian optimization tends to focus on complexity analysis in terms of iteration rounds, instead of asymptotic behavior. I was wondering if it would be possible to extend your proposed RRM framework to a non-asymptotic setting. In other words, can the avoidance probability be expressed in terms of the number of iteration rounds, like some approaches in Euclidean space? If this is not feasible, could you elaborate on the main hindrances or challenges in doing so?

Ref: [1] J LI et.al., Stochastic Zeroth-order Riemannian Derivative Estimation and Optimization
[2] X Wang et.al., Online Optimization over Riemannian Manifolds

**Limitations:**

The authors have outlined the limitations of their work.

---

> ### Author Rebuttal · Authors · 2023-08-08
>
> Thank you for your input and remarks. We reply to your questions below, and we will revise our manuscript accordingly in the next revision opportunity.
>
>
> > It would be better if there are numerical experiments to validate the findings.
>
> Done - please see the "global rebuttal" where we included a series of plots on the torus to illustrate how RSGD and RSEG avoid saddle points, despite being initialized closed to spurious saddle points. [The plots are all in the the one-page pdf that we were allowed to upload as part of our rebuttal.]
>
>
> >In the conclusion section of your paper, you mention 'zeroth-order optimization' as a potential avenue for future work. There have been several studies related to the zeroth-order (or so-called 'bandit') gradient descent method ([1,2]). I am interested to know whether such zeroth-order methods fit within your proposed Riemannian Robbins-Monro (RRM) scheme. If not, could you please clarify the primary gap or obstacle in incorporating these methods into the RRM framework?
>
> This question is fairly intricate:
>
> 1. It is not hard to show that the methods in the provided references, as well as a number of other zeroth-order Riemannian methods, indeed fall under the RRM template with certain explicit expressions on the noise and bias/offset terms ($U_n$ and $b_n$ respectively).
>
> 2. However, a simple calculation reveals that
> $$
> \|U_n\| = O(\mu_n^{-2}), \quad  ||b_n || = O(\mu_n).
> $$
> where, in line with the notation in reference [1] that you provided, $\mu_n$ is the *sampling radius* of the zeroth-order scheme under scrutiny (also known as smoothing parameter, exploration parameter, etc.). In zeroth-order methods, this parameter typically goes to $0$ as $n\to\infty$ to enable convergence so, even though the bias/offset assumption in (12) is not violated, the noise bound becomes time-varying, and our results do not immediately apply in this context.
>
> Addressing these challenges, potentially through a careful balance between the step-sizes $\gamma_n$ and $\mu_n$ (plus a suitable extension of the probabilistic estimates of Lemma B.1), is an intriguing research question - but, at the same time, it would require orthogonal work to the current paper, so we left it as a direction for future research.
>
>
> >A lot of work on Riemannian optimization tends to focus on complexity analysis in terms of iteration rounds, instead of asymptotic behavior. I was wondering if it would be possible to extend your proposed RRM framework to a non-asymptotic setting. In other words, can the avoidance probability be expressed in terms of the number of iteration rounds, like some approaches in Euclidean space? If this is not feasible, could you elaborate on the main hindrances or challenges in doing so?
>
> It is certainly possible to enhance our analysis further through the imposition of more stringent conditions, such as the capability to intentionally introduce artificial noise, as discussed in references like [20,63], or additional structure on the function being optimized - e.g., like the "$(\alpha,\beta,\gamma,\delta)$-strict-saddle" property of Ge et al. [22] and Jin et al. [25]. In this more restrictive context, we believe it should be possible to obtain a result along the lines that you suggest - i.e., providing an upper bound on the number of iterations required to produce "an" iterate which is sufficiently far from a saddle point. However, this would not imply avoidance, but a "best iterate with high probability" guarantee - i.e., in principle, the sequence of iterates could still converge to a strict saddle, even if, with high probability, it has produced an iterate which is close to a local minimum. [We should perhaps also emphasize at this point that this phenomenon is not exclusive to the manifold context, as similar scenarios are encountered in Euclidean optimization problems.]
>
> ---
>
> We hope and trust that these points address your questions - but please let us know if any of the above is not sufficiently clear.
>
> Thank you again for your input and positive evaluation,
>
> The authors

---

> > ### Comment · Reviewer_5i8d · 2023-08-12
> >
> > Thank you for addressing my concerns. In light of your response, I have increased my score from 6 to 7.

---

> > > ### Author Response · Authors · 2023-08-15
> > >
> > > Thank you for your input and upgraded re-assessment!

---

### Official Review · Reviewer_MfPv · 2023-07-09

**Soundness:** 4 excellent
**Presentation:** 4 excellent
**Contribution:** 4 excellent
**Rating:** 6
**Confidence:** 4

**Summary:**

This paper studies the problem of when Riemannian first-order optimization algorithms evade strict saddles points. The proof leverages a connection to the Euclidean case, to the continuous-time Riemannian gradient flow, and shows that the deviation from them is bounded. Intuitions are given in the main paper to facilitate readership.

**Strengths:**

The paper has done a great job in writing, connecting with existing literature, and giving intuitions on how the proof is done. I agree with the claimed contribution in Lines 42-48 and 276-292.

**Weaknesses:**

I don't see major problems. I do have a few questions and minor comments below.

## Questions
1. For the definition of a stable manifold: why do we need positive eigenvalues lower bounded by c+>0? What goes bad if the positive eigenvalues could get arbitrarily close to 0?
1. Equation 6: Consider any update rule whose output is in the range of the exponential map. We can cast the update as (RRM), as is also done in (6). I understand that the gain is that we can then analyze just the exponential map such as the arguments around (15). However, the difficulty is then transferred to checking whether the logarithm of the update satisfies assumption 2, which is much more indirect than checking whether V satisfies assumption 2 as in Algorithm 1. Is this a problem? Could you comment more on this?
1. Related to the above, for algorithm 3, why do I view it as a special case of algorithm 2 and hence of RRM, rather than saying oh, algorithm 3 is a special case of RRM, because one can just take logarithm and exponential map of (SMD)?
1. Does (RGF) always stay within the manifold? If so, why? If not, doesn’t the ‘grad’ become not defined outside the manifold?

## Comments
1. Line 124-125: The definition looks recursive hence confusing: to understand vn_hat I need to understand Un, bn, both of which depend on vn_hat.
1. Line 18: “end state of a stochastic Riemannian algorithm can only be a local minimizer.” Do you mean something like the limiting state? Even for convex problems in Euclidean space, gradient descent is only guaranteed to approach a solution - the more iterations, the closer. So, in general, the end state is something close to a local minimizer.
1. Line 102: the “S” is not defined - I suppose you mean a stable manifold?




**Questions:**

See above

**Limitations:**

Yes

---

> ### Author Rebuttal · Authors · 2023-08-08
>
> Thank you again for your input and remarks. We reply to your questions below, and we will revise our manuscript accordingly in the upcoming revision.
>
> >For the definition of a stable manifold: why do we need positive eigenvalues lower bounded by c+>0? What goes bad if the positive eigenvalues could get arbitrarily close to 0?
>
> What is really needed is to be able to invoke the stable manifold theorem. In this regard, the $c_{+}$ lower bound is simply a convenient way to ensure that the positive, zero and negative eigenspaces of the Hessian of $f$ do not change dimension across the saddle manifold, and they induce a well-defined separation of the tangent bundle to $S$ (as a sub-bundle of the tangent space of $M$). It would be possible to relax this condition with no impact on our results, but this is the most straightforward condition that we're aware of in the literature.
>
>
> >Equation 6: Consider any update rule whose output is in the range of the exponential map. We can cast the update as (RRM), as is also done in (6). I understand that the gain is that we can then analyze just the exponential map such as the arguments around (15). However, the difficulty is then transferred to checking whether the logarithm of the update satisfies assumption 2, which is much more indirect than checking whether V satisfies assumption 2 as in Algorithm 1. Is this a problem? Could you comment more on this?
>
> Indeed, if an algorithm has a "look-ahead" structure (potentially involving a parallel transport step, like RSEG or the optimistic version of RSGD), the verification of Assumption 2 might require some work, and it is not an immediate consequence of the oracle / stochastic gradient assumptions. So, yes, in theory, if one seeks to employ the RRM framework for a given algorithm, they would have to verify whether Assumption 2 is verified for said algorithm. In practice however, in all the examples that we tested from the literature (Algorithms 1-7, and several that we did not include to avoid overbloating the presentation), this verification is a relatively straightforward affair that did not present any major difficulties (conceptual or technical).
>
>
> >Related to the above, for algorithm 3, why do I view it as a special case of algorithm 2 and hence of RRM, rather than saying oh, algorithm 3 is a special case of RRM, because one can just take logarithm and exponential map of (SMD)?
>
> This is indeed related to your previous point. In the case of Bregman-based methods, the main difficulty lies in the verification of **Assumption 2**, which explains our preference for viewing (SMD) as a special case of retraction-based methods, as opposed to a direct adoption of the logarithm map.
>
>
> >Does (RGF) always stay within the manifold? If so, why? If not, doesn’t the ‘grad’ become not defined outside the manifold?
>
> Yes, $\mathrm{grad} f$ is a section of the manifold's tangent bundle so, by completeness, RGF gives rise to a global flow on the manifold, and hence always remains there, cf. [36, Chaps. 9 and 10].
>
>
> >Line 124-125: The definition looks recursive hence confusing: to understand vn_hat I need to understand Un, bn, both of which depend on vn_hat.
>
> Whoops, yes, Line (125) was supposed to be a consequence of (124). We will fix this, apologies for any confusion.
>
>
> >Line 18: “end state of a stochastic Riemannian algorithm can only be a local minimizer.” Do you mean something like the limiting state? Even for convex problems in Euclidean space, gradient descent is only guaranteed to approach a solution - the more iterations, the closer. So, in general, the end state is something close to a local minimizer.
>
> Yes, by "end state" we meant the "limit state", not that a minimizer can be reached in a finite number of steps (in general, this is not possible). Consider this fixed in our revision.
>
> ---
>
> We hope that the above addresses your questions - but please let us know if any of the above is not sufficiently clear.
>
> Thank you again for your input and positive evaluation,
>
> The authors

---

### Author Rebuttal · Authors · 2023-08-09

Dear AC, dear reviewers,

We are sincerely grateful for your time, input and positive evaluation.

To streamline our rebuttal, we reply to each reviewer's questions in a separate point-by-point thread below. We only include in this global rebuttal a pdf with two figures showing the avoidance of saddle points under the RSGD and RSEG methods (Riemannian stochastic gradient and extra-gradient respectively). For illustration purposes, we used a $2$-dimensional torus and an objective function with three saddle points and one minimizer, depicted as black and red respectively. Despite being initialized close to the spurious saddle points, both methods avoid them and ultimately converge to a (global) minimum, as suggested by the theory of the paper.

We defer all other points to the reviewer-specific threads below and we are looking forward to the discussion phase if any further questions remain.

With our kindest regards,

The author team of Paper 6491

---

### Decision · Program_Chairs · 2023-09-21

**Decision:**

Accept (poster)

**Comment:**

This paper answers an important question: when can Riemannian stochastic algorithms avoid saddle point? This question has been addressed in the Euclidean space and for the Riemannian deterministic algorithm. Therefore, the results provide important complement to the literature. The results of this paper are timely and valuable since both Riemannian optimization and stochastic algorithms are becoming more and more important and find many important applications. The paper is well-written and organized.